# The Impact of the COVID-19 Pandemic on Burnout, Compassion Fatigue, and Compassion Satisfaction in Healthcare Personnel: A Systematic Review of the Literature Published during the First Year of the Pandemic

**DOI:** 10.3390/healthcare10020364

**Published:** 2022-02-13

**Authors:** Cristina Lluch, Laura Galiana, Pablo Doménech, Noemí Sansó

**Affiliations:** 1Department of Methodology for the Behavioral Sciences, University of Valencia, 46010 Valencia, Spain; cris9@alumni.uv.es (C.L.); laura.galiana@uv.es (L.G.); 2Faculty of Education, Valencian International University, 46002 Valencia, Spain; pablo.domenech@campusviu.es; 3Department of Nursing and Physiotherapy, University of Balearic Islands, 07122 Palma, Spain; 4Balearic Islands Health Research Institute (IDISBA), 07120 Palma, Spain

**Keywords:** COVID-19, burnout, compassion fatigue, compassion satisfaction, health personnel

## Abstract

This literature review aimed to determine the level of burnout, compassion fatigue, and compassion satisfaction, as well as their associated risks and protective factors, in healthcare professionals during the first year of the COVID-19 pandemic. We reviewed 2858 records obtained from the CINAHL, Cochrane Library, Embase, PsycINFO, PubMed, and Web of Science databases, and finally included 76 in this review. The main results we found showed an increase in the rate of burnout, dimensions of emotional exhaustion, depersonalization, and compassion fatigue; a reduction in personal accomplishment; and levels of compassion satisfaction similar to those before the pandemic. The main risk factors associated with burnout were anxiety, depression, and insomnia, along with some sociodemographic variables such as being a woman or a nurse or working directly with COVID-19 patients. Comparable results were found for compassion fatigue, but information regarding compassion satisfaction was lacking. The main protective factors were resilience and social support.

## 1. Introduction

Because healthcare professionals are especially exposed at the frontline of the COVID-19 pandemic, their quality of life has been put at great risk. Among several potentially harmful factors for the health of professionals, some authors have highlighted the lack of access to adequate protective equipment [1], exhaustion resulting from wearing personal protective equipment throughout the working day, the feeling of having inadequate support [2,3], long working hours and unexpected changes in the type of work [4], concern about trapping or infecting their relatives [5], abandoning their homes to avoid infecting their families [6], lack of access to updated information on constantly changing patterns of action [3,7], uncertainty about disease containment [1], and concerns about seeing patients die [5]. 

Thus, it seems clear that health professionals are under extreme psychological pressure and, consequently, are at risk of developing several psychological symptoms and mental health disorders [3]. For example, a recent review that included data from more than 7000 professionals [4] found that the prevalence of PTSD symptoms and anxiety and depression ranged from 9.6% to 51% and 20% to 75%, respectively. High levels of stress and somatic symptoms were also reported in Italian health professionals in the study by Barello et al. [8]. Furthermore, a study by Kotera et al. [9] in Japan, found that physicians had more mental health disorders, felt more alone, and had less hope and self-compassion compared to the general population.

Therefore, it is not surprising that the COVID-19 pandemic has worsened the quality of life of professionals, aggravating pre-existing problems such as burnout. Burnout, or professional burnout, is a syndrome that occurs in service sector workers subjected to stressful situations [10], and can be defined as the “result of chronic stress in the workplace that has not been successfully managed” [11]. The academic literature from the past few decades has revealed that health professionals are especially vulnerable to burnout because their work context is characterized by high-risk decisions, dealing with the public, and expectations of compassion and sensitivity [12]. However, more and more academics and clinicians have pointed out that burnout alone is insufficient to explain the emotional problems presented by practitioners in general healthcare contexts [13,14] or in the field of palliative care, in particular [15,16]. In this sense, a large body of recent evidence now suggests that many healthcare professionals are suffering from compassion fatigue [17,18].

The concept of compassion fatigue was first introduced by Joinson [19] to characterize a state of reduced capacity for compassion as a consequence of exhaustion caused by contact with the suffering of others [20]. Witnessing the suffering of patients without being able to alleviate their discomfort has a high emotional toll on healthcare personnel [21]. The most widespread theoretical model for the study of compassion fatigue is currently the one developed by Beth Stamm [22,23], who defined it as the negative aspect of professional quality of life and divided it into two dimensions: (1) burnout (as previously explained), and (2) secondary trauma, vicarious trauma, or secondary traumatic stress, which refers to negative feelings driven by fear and trauma related to work [23]. This model also studies the opposite pole of compassion fatigue, that is, the positive aspects of professional life, or compassion satisfaction. 

Compassion satisfaction occurs when exposure to traumatic and distress-related events produces satisfaction [24] derived from the pleasure of helping others [22] and providing a means to alleviate suffering [24]. Indeed, when helping people and changing their lives is managed appropriately, professionals and caregivers can feel pleasure and satisfaction rather than burnout or compassion fatigue [14]. Therefore, considering the definition of professional quality of life, it seems clear that the circumstances created by the COVID-19 pandemic are a clear threat to the mental health of professionals and may have affected their levels of burnout, compassion fatigue, and compassion satisfaction. In turn, these factors are key to achieving the adequate well-being of healthcare providers [25,26,27,28] and, in turn, can strongly affect the quality of care received by patients and their families [29].

Although several systematic reviews have recently been published on the impact of the COVID-19 pandemic on the mental health of healthcare professionals [4,30,31], very few of them specifically included dimensions of professional quality of life such as burnout [32,33,34,35], only one considered indirect trauma [36] and, to the best of our knowledge, none reviewed the evidence on compassion satisfaction. In this context, the main objective of this current work was to understand the impact of the COVID-19 pandemic on burnout, compassion fatigue, and compassion satisfaction among health professionals by systematically reviewing the literature published during the first year of the COVID-19 pandemic. Specifically, we aimed to answer the following questions, all of them referring to the experience lived during the first year of the COVID-19 pandemic:What levels of burnout, compassion fatigue, and compassion satisfaction have health professionals who worked during the COVID-19 pandemic experienced?What variables (risk factors) were related to the COVID-19 pandemic having a greater negative impact on professional quality of life?What variables (protective factors) corresponded to the COVID-19 pandemic having a lower negative impact on professional quality of life?

## 2. Materials and Methods

To complete this work we conducted a systematic review of the scientific literature, following the Preferred Reporting Items for Systematic Review and Meta-analysis (PRISMA) guidelines [37] (see Appendix A).

### 2.1. Eligibility Criteria

#### 2.1.1. Type of Participants

Health professionals (physicians, nurses, nursing assistants, psychologists, etc.) who carried out their professional activities in the health system (such as primary healthcare centers, emergency departments, intensive care units, palliative care units, or COVID-19 services, among others) during the COVID-19 pandemic were considered in this work.

#### 2.1.2. Study Variables

We considered studies that addressed burnout, compassion fatigue, and compassion satisfaction in health professionals who had cared for patients infected by COVID-19. 

#### 2.1.3. Study Types

We included quantitative studies (either cross-sectional or longitudinal) with primary data that addressed burnout, compassion fatigue, and compassion satisfaction in healthcare professionals during the COVID-19 pandemic. Studies published, or that were in press, from 1 January 2020 to 31 December 2020, were considered.

#### 2.1.4. Language

Articles published in English or Spanish were included. 

#### 2.1.5. Publication Date

Articles published during 2020 (from 1 January 2020 to 31 December 2020) were considered.

#### 2.1.6. Exclusion Criteria

The following types of work were excluded: studies that did not consider healthcare professionals; did not include our study variables (burnout, compassion fatigue, and/or compassion satisfaction); did not include quantitative primary data (i.e., single case studies, reviews, letters to the Editor, comments, qualitative studies, etc.); were not published in Spanish or English; were not published during the year 2020; and that included data from before the COVID-19 pandemic, even when the work met all the inclusion criteria. 

### 2.2. Data Sources and Search Strategy

We searched the CINAHL, Cochrane Library, Embase, PsycINFO, PubMed, and Web of Science databases for relevant articles. Thus, we only used reliable, peer-reviewed databases, platforms, and sources with search tools that allowed us to access the study dates, and thereby, systematically identify studies. These databases included academic literature related to various health disciplines, including health psychology, and therefore represented reliable sources of expert research and information. 

The keywords we used were:*Pandemic* or *COVID-19* or *SARS-CoV-2* or *Coronavirus*, as well as the synonyms for these terms included in the Medical Subject Headings (MeSH) database; and*Burnout* or *compassion fatigue* or *stress disorders* or *compassion satisfaction*, as well as the synonyms for these terms included in the MeSH; and*Health personnel* or *nursing staff* or *nurses* or *physicians* or *psychology*, as well as the synonyms for these terms included in the MeSH.

A list of the terms found in the MeSH, together with the equation we used in the final search, is provided in Appendix A.

Regarding the review procedure, first we entered the search equation into each of the databases, filtering them by publication date (1 January 2020 to 31 December 2020) to narrow the results based on the eligibility criteria. Second, the eligible papers were identified based on their title and keywords as well as whether they met the inclusion criteria. Third, we read the abstracts, reserving any studies we believed met the inclusion criteria. Finally, the full texts of these articles were obtained and read. After this reading, we chose the final records for inclusion and performed the data synthesis.

### 2.3. Data Extraction and Synthesis

The data from the publications obtained in the search strategy were extracted into an Excel template that was modified according to the studies we reviewed. The metadata included the author(s), year of publication, country of study, main study objective, design, sample size, types of participating professionals, distribution by gender, mean age, other sample characteristics, assessment instruments, metrics used for each variable, descriptive and inferential results relative to the prevalence, data collection date, risk factors and protective variables for burnout, compassion fatigue, and compassion satisfaction. Specifically:Means and standard or median deviations and interquartile ranges (for quantitative data), frequencies and percentages (for categorical data) of the prevalence data for burnout, compassion fatigue, and compassion satisfaction.To study the risk factors and protective variables of burnout, compassion fatigue, and compassion satisfaction, chi-squared tests, contrast of means, and analysis of variance (for categorical variables), Pearson correlations, Spearman correlations, and simple and multiple regressions (for quantitative variables) were used.

## 3. Results

When applying the inclusion and exclusion criteria to the results from the six databases, the search equation produced 2856 records. As shown in Figure 1, these were reduced to 2498 records once the publication date was limited. We reviewed all these entries, first by title and then by abstract, leaving a total of 234 total records for full text review. Most of these were excluded because they did not meet one or more of the inclusion criteria (i.e., health professional participants, burnout, compassion fatigue, compassion satisfaction variables, etc.). After reading the full texts of all these entries, 76 records were retained for inclusion in this review. The main characteristics of these tests are summarized in Table 1.

The research included in this review was all carried out between February and May 2020, with most of the studies having collected data between March and April (that is, Kannampallil et al. [72]; Ruiz-Fernández et al. [95]; Trumello et al. [101]). Most of the study samples included 100 to 400 participants; Chen et al. [48] included the largest number of participants (12,596 people), while the smallest study cohort was limited to 80 participants [59]. As shown in Table 1, physicians and nurses were the most-studied groups, either separately or together, during the health crisis caused by SARS-CoV-2. In addition, several researchers also focused on other medical professionals including residents, assistants, administrative personnel, physiotherapists, and laboratory technicians, among others. In terms of gender, the samples in 83.4% of the studies comprised more than 50% women, with only 16.4% of the articles including more men than women [39,41,42,50,51,52,54,56,64,74,89,94].

The impact of the COVID-19 pandemic on health professionals of different nationalities was also studied. The most-examined country was the United States of America, with 15 studies [38,49,50,52,56,65,72,73,75,81,86,92,94,104,109]. The two European countries in which the effects of COVID-19 were most studied were Italy and Spain, with eleven [8,43,45,55,57,66,67,68,93,101,102] and seven articles [44,62,78,82,83,84,95,98], respectively.

Regarding the study variables, burnout was studied in 67 (88%) of the papers included in this review, most often with the Maslach Burnout Inventory [8,40,42,43,46,48,54,55,56,57,58,60,68,70,78,84,85,87,88,89,92,93,96,97,104,105,106,107,109,110]. A total of 61% of the studies used the aforementioned questionnaire or one of its derivatives: the aMBI [52,74,75], CMBI [7,48], Mini-z MBI [50,51], MBI-HHS [39,71,82,90,108], or PWLS [73]. Other authors developed an ad-hoc questionnaire [94,103] or used instruments such as the CBI [49,63,66,67,76], OLBI [53,65,69,100], ProQOL [45,62], PFI [61,72,109] or CESQT [83,98]. The highest burnout found in the reviewed studies was been for infectious disease physicians in the Republic of Korea, with 90% of them presenting burnout [90]. The lowest burnout was found in a study carried out in Spain, in which burnout was present in 20.4% of health professionals [84].

The average level of burnout among healthcare professionals was high, especially on the emotional exhaustion and depersonalization subscales [7,39,43,52,63,64,71,88]. Some studies indicated high scores as a consequence of the pandemic on the personal accomplishment subscale [75,78], although these were lower in other studies [7,39,52,60,84]. Numerous reports pointed out the influence that some variables had on the perception of burnout, although 31% (24) of the studies that evaluated burnout did not study its relationship with other variables. The most-studied variables were gender, profession, and workplace (COVID-19/frontline rooms vs. non-COVID-19/secondline rooms). Women showed higher scores on the emotional exhaustion and depersonalization subscales [43,51,76,85,90,93,96,103,110]. 

Regarding the professional category, higher burnout scores were reported for nurses in several articles [46,49,65,85,100,103], although others pointed towards higher levels of burnout among physicians [62,94]. In terms of the workplace, the results were also contradictory; some research indicated that health workers on the frontline against COVID-19 suffered less burnout [58,104], while the majority found higher burnout scores among these same health workers [40,63,69,72,93,95,101,106,109]. Compared to the general population, healthcare personnel showed higher burnout scores [55,80]. The number of patients attended to also appeared to influence the level of exhaustion: the more COVID-19 patients seen by the participants, the higher their levels of exhaustion [54].

The risk factors, or those that showed a positive relationship with burnout, were anxiety and depression [82,106,108], insomnia [96,106], and moral damage [106]. Work stress also influenced burnout [69] and the lack of personal protective equipment affected emotional exhaustion [39,57]. Protective factors, or those whose presence was related to lower levels of burnout, included resilience and social support [70], and quality of life [46]. In addition, two studies observed that different interventions positively affected the levels of burnout in health workers. For example, Dincer and Inangil [59] implemented a program of emotional freedom techniques that reduced the level of burnout in healthcare personnel. Likewise, Lee et al. [79] found that following a coping strategies program resulted in lower levels of burnout among healthcare professionals. Finally, positive correlations were observed between burnout and secondary trauma or compassion fatigue [45].

Compassion fatigue was also studied in 16 (21%) of the studies included in the review. Of note, some studies referred to the concept as compassion fatigue [62,66,91,95,101,109] while others referred to it as secondary or vicarious trauma [17,38,44,45,67,77,80,97,101,102]. The instruments most used to assess these variables were the ProQOL-5 [45,62,66,81,91,95,101] and STSS [17,97,102]. The levels of compassion fatigue or vicarious trauma found in healthcare professionals were generally high [17,44,45,91,101], although in specific studies they were medium [38] or low [62].

Regarding the protective and risk factors for compassion fatigue, again, the studies we considered focused on variables such as gender, profession, or workplace. Specifically, working with COVID-19 patients tended to increase secondary trauma scores [44,109]. Being a woman was also associated with higher levels of compassion fatigue [17,91]. Additionally, the professional category seemed to influence the perception of fatigue, although the results were inconclusive. Physicians showed higher compassion fatigue scores in the study by Ruiz-Fernández et al. [95]. Franza et al. [67] found that mental health workers had higher compassion fatigue scores, while the groups of therapists and nurses showed reduced compassion fatigue and lower scores on the burnout and secondary trauma subscales with respect to groups of physicians and psychologists. In any case, studies of this nature were scarce. 

Finally, compassion satisfaction was only studied in four (5%) of the studies included in this current review [45,62,91,95] and the ProQOOL-5 questionnaire [23] was used to assess this factor in all these studies. In terms of the levels of compassion satisfaction, the study by Buselli et al. [45] found mean levels of 38.2 ± 7.0 for the sample of physicians and nurses. Along the same lines, Dosil et al. [62], reported high (33.2%) or medium (63.1%) levels of compassion satisfaction in health professionals. The latter authors also observed a relationship between compassion satisfaction and professional category, with higher levels of compassion satisfaction being reported in medical assistants/technicians compared to nurses and physicians. In contrast, Ruiz-Fernandez et al. [95] found that nurses had higher scores for compassion satisfaction than physicians. The last study that evaluated compassion satisfaction did not provide descriptive or inferential data in this regard [91].

## 4. Discussion

The main objective of this work was to understand the impact of the COVID-19 pandemic on the quality of life of healthcare professionals, specifically in terms of burnout, compassion fatigue, and compassion satisfaction. To this end, we carried out a systematic review of the literature produced during the first year of the pandemic (2020) which, after screening 2856 records, finally included 76 research papers. The main characteristics of the samples included in the reviewed studies agreed with those previously reported for health professionals in other systematic reviews, in which nurses and women predominated as the main participants [111,112,113]. Considering the results we obtained, it is evident that burnout is still used as the main indicator of emotional well-being in health professionals, much more so than other variables such as fatigue and compassion satisfaction that are used in the more recent literature. The data in this review coincided with those from Mol et al. [113], in which 88% of the articles they examined evaluated exhaustion, while compassion fatigue and compassion satisfaction was considered only in 21% and 5% of the cases, respectively. 

Regarding the effect of the pandemic on health workers, we observed a worsening of the level of burnout. Specifically, in many of the studies [7,39,43,52,63,64,71,88], the scores for emotional exhaustion and depersonalization exceeded the medium–high levels obtained in pre-pandemic reviews [111,114,115]. However, for certain professional profiles such as healthcare professionals in the oncology area, elevated levels of emotional exhaustion and depersonalization had already been identified prior to the pandemic [116]. Of note in this present review was the fact that, compared to previous reviews which described very heterogeneous prevalences ranging from 0% to 80% [117], the levels of burnout in the studies we considered were more homogeneous, ranging from 30% to 60% [39,43,55,71].

In the same way that burnout increased in health professionals during the COVID-19 pandemic, an increase in compassion fatigue or vicarious trauma was also observed during the same period. Coinciding with the review by Xie et al. [118] implemented in emergency nurses before the COVID-19 pandemic, our results suggest that healthcare professionals had high scores for compassion fatigue [17,44,45,91,95,101], although other studies [38] have reported medium levels for this factor. In contrast, reviews conducted prior to the COVID-19 pandemic found moderate levels of compassion fatigue among healthcare workers [114,115]. However, very little data regarding compassion satisfaction were available, and some of these studies were inconclusive [91]. The levels of compassion satisfaction were generally medium or high [62], and were similar to those from before the COVID-19 pandemic. For example, the data collected by Xie et al. [118] from 2015 to 2019, found medium compassion satisfaction levels among oncology nurses.

Regarding the risk factors for developing burnout, our results indicate that the variables that influenced professional quality of life were gender (female sex), profession (nursing), and the workplace (attending or not attending patients with COVID-19). Indeed, the first two risk factors have already been recorded elsewhere in the literature [111,119]. Other variables that emerged as risk factors in this current review included anxiety, depression, and insomnia. Along the same lines, Gómez-Urquiza et al. [116] also highlighted these same risk factors for burnout. Furthermore, the same risk factors have also been observed for compassion fatigue. Finally, given the scarcity of results, we were unable to detect risk or protective factors for compassion satisfaction. In agreement with results from before the pandemic, the protective factors against burnout detected in this review included resilience, social support, and participating in interventions to reduce burnout. For example, in research on resilience and burnout, Heath et al. [120] found that preventive strategies, self-care, organizational justice, and having individual and organizational preventive strategies were protective factors against the development of emotional exhaustion. In fact, these interventions were already being implemented, and were equally effective before the COVID-19 pandemic [113].

Finally, it is worth highlighting both the strengths and limitations of this present study. Of note, although some literature reviews from prior to the pandemic focused on burnout among healthcare professionals, very few studies evaluated compassion fatigue, and even fewer studied the effect of compassion satisfaction. Reviews that focused on the impact of COVID-19 were much scarcer, with only one considering compassion fatigue and none having reviewed the literature on compassion satisfaction. Additionally, even though research focusing on burnout was more abundant, a much higher proportion of the relevant academic literature was considered in this present review. For example, the review by Sharifi et al. [35] on burnout among healthcare workers during the COVID-19 pandemic only included 12 studies; Chew et al. considered 23 studies evaluating emotional exhaustion and other variables; and Amanullah and Ramesh [32] only included five articles. Similarly, the only review available on vicarious trauma only included seven studies [36] compared to the 16 considered in this present review. Regarding the limitations of our work, we did not assess the quality of the articles we included in this review. Furthermore, to facilitate the synthesis of the results, we only included quantitative studies; therefore we may have excluded qualitative studies containing relevant information. In this sense, future work could assess the information collected in these qualitative studies.

## 5. Conclusions

Based on the results of our work, and in light of the literature we reviewed, we concluded that the quality of life of health professionals was significantly affected by the COVID-19 pandemic. Specifically, burnout levels increased from medium–high to high and compassion fatigue went from medium to high. Healthcare professionals reported high rates of emotional exhaustion, depersonalization, low personal accomplishment, and compassion fatigue, and low rates of compassion satisfaction. In addition, given that research from all five continents was included in this review, these findings can be considered global. 

Therefore, in light of the results, we can say that the vulnerability of healthcare professionals to processes such as burnout or compassion fatigue has increased even more as a result of the COVID-19 pandemic. These problems, in addition to affecting the quality of care provided by health personnel, have a negative impact on professionals’ well-being and quality of life; this can aggravate the lack of health professionals that health systems have suffered around the world for decades. Of note, the risk factors and protective factors have not changed compared to previous findings, meaning that we already have the scientific knowledge required to implement interventions to mitigate the empathic burnout of our healthcare professionals. We can no longer afford not to implement preventive strategies to prevent processes such as burnout and compassion fatigue in health professionals.

## Figures and Tables

**Figure 1 healthcare-10-00364-f001:**
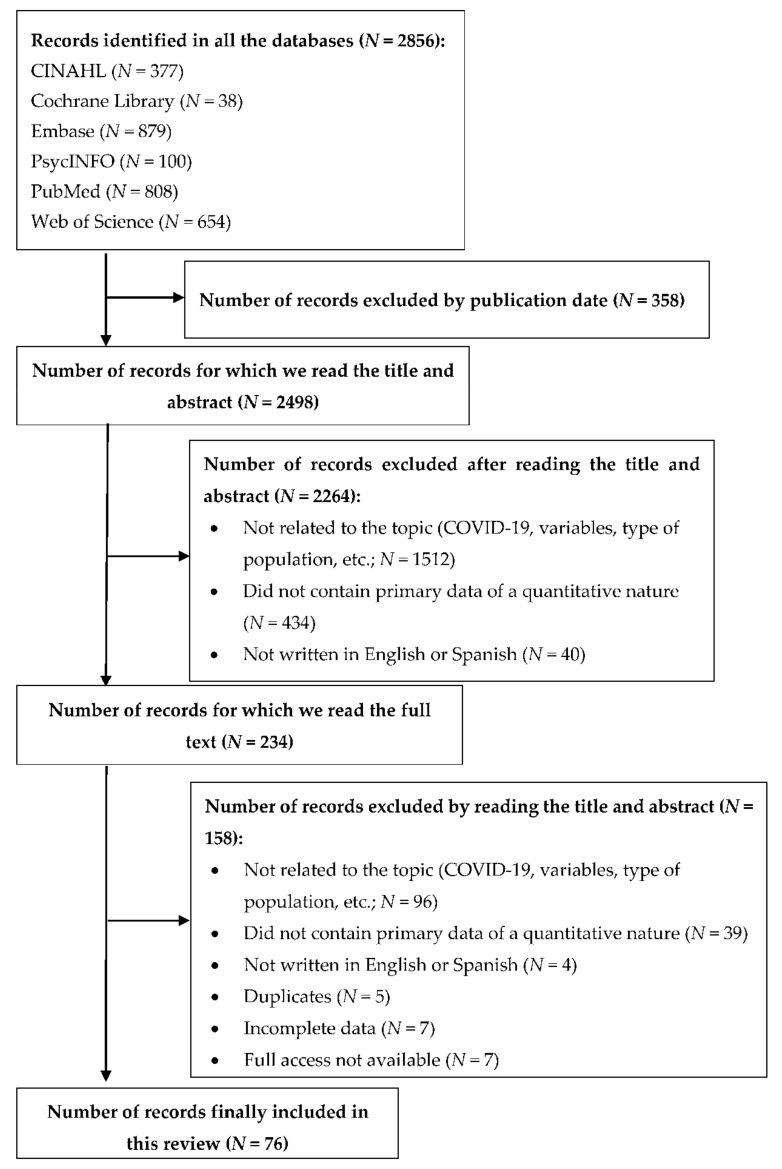
Article selection flow chart.

**Table 1 healthcare-10-00364-t001:** The main characteristics of the studies included in this systematic review.

Author and Year	Aims	Country	Sample	Results
Aafjes-van Doorn (2020) [38]	To learn how vicarious trauma affected psychotherapists during the COVID-19 pandemic.	United States of America	*n* = 339 psychotherapists Men: 90 Women: 249	Vicarious trauma (VTS): The mean level of moderate vicarious trauma was *M* = 33.46, 8.5%. Vicarious trauma was greater in therapists who experienced more distress in response to what they had heard in sessions during the pandemic compared to previous sessions (*r* = 0.22, *p* < 0.001). They felt more tired (*r* = 0.16, *p* < 0.010), less competent (*r* = 0.21, *p* < 0.001), and less confident (*r* = 0.15, *p* < 0.010) in their therapy sessions compared to their pre-pandemic feelings. They also reported a deterioration in their therapeutic work: less emotional connection with their clients (*r* = 0.18, *p* < 0.001), and a weaker therapeutic relationship (*r* = 0.16, *p* < 0.010) post-pandemic.
Abdelhafiz et al. (2020) [39]	To assess the prevalence of burnout in Egyptian physicians during the COVID-19 pandemic and study the associated risk factors and possible consequences.	Egypt	*n* = 220 Specialties: Thorax: 75 Anesthesia and intensive care: 32 Internal medicine: 21 Family doctor: 11 Surgery: 11 Radiology: 8 Tropical medicine: 6 Pediatrics: 5 Emergency doctors: 4 Clinical pathology: 3 Cardiology: 3 Others: 41 Men: 111 Women: 109	Burnout (MBI-HSS): 28.18% presented severe exhaustion, 31.82% severe depersonalization, and 89.09% a severe decline in personal accomplishment. The lowest personal accomplishment scores were correlated with dissatisfaction with patient cure rates (*p* = 0.040) and the death of a colleague or family member from COVID-19 (*p* = 0.040). The highest scores for emotional exhaustion (*p* < 0.001) and depersonalization (*p* = 0.004) were associated with the need to buy personal protective equipment (PPE) with personal money, and harassment by patient families while working (*p* < 0.001 for both). Emotional exhaustion was associated with the absence of PPE in the workplace (*p* = 0.040), a lack of public awareness of the disease (*p* = 0.010), insufficient public appreciation of the work undertaken by physicians during the pandemic (*p* = 0.010), and dissatisfaction with their salaries (*p* = 0.020).
Aebischer et al. (2020) [40]	To compare the physical and psychological health of Swiss medical students involved in the COVID-19 response with that of their non-involved peers. To compare the health of those who work on the frontline with that of peers not on the frontline. To compare frontline medical students with frontline medical residents.	Switzerland	*n* = 777 Students involved in COVID-19: 296 Students not involved in COVID-19: 254 Medical residents: 227 Men: 250 Women: 527	Burnout (MBI), generalized anxiety (GAD-7), and depression (PHQ-9): Students involved in the response to COVID-19 reported a similar proportion of COVID-19 symptoms or confirmed diagnoses (*p* = 0.810), but lower levels of anxiety (*p* < 0.001), depression (*p* < 0.001), and depersonalization (*p* < 0.001), compared to their non-involved peers. The health of frontline students did not differ significantly from those who were not frontline. Frontline students had lower levels of emotional exhaustion than frontline residents (*p* < 0.010).
Jha et al. (2020) [41]	Characterize the degree of burnout and specific characteristics related to the COVID-19 pandemic.	United States of America	*n* = 100 interventional pain physicians Men: 81 (81%) Women: 19 (19%)	Burnout: In response to the specific indicator, “Are you feeling exhausted right now?”, participants responded with yes/no: yes: 52 (52%); no: 46 (46%); unanswered: 2 (2%). Most reported that COVID-19 had impacted their operation (main modifications: telemedicine and interruption of procedures [98%]; a reduction in earnings [91%]; and reduced spending due to uncertainty [77%]).
Arpacioglu et al. (2020) [17]	To study the differences in secondary trauma experienced by frontline workers and by the general population.	Turkey	*n* = 563 Health workers: 251 Physicians:124 Nurses: 93 General population (non-health workers): 95 Men: 212 (37.7%) Women: 351 (62.3%)	Secondary trauma (STSS) and anxiety and depression (PHQ-4): Health workers who worked with COVID-19 patients had the highest secondary trauma score (*M* = 2.66 ± 0.96) while the general population obtained the lowest score (*M* = 2.34 ± 0.76). Anxiety/depression levels: health workers working with COVID-19 patients (*M* = 2.25 ± 0.79), those not working with COVID-19 patients (*M* = 2.01 ± 0.70), non-health workers (*M* = 1.82 ± 0.59). Women presented higher compassion fatigue (*p* = 0.048), which was significantly higher in nurses (*p* = 0.004) and physicians (*p* = 0.022) compared to the general population. Anxiety was significantly higher in nurses compared to other health professionals (*p* = 0.038).
Azoulay et al. (2020) [42]	Document the prevalence of mental health outcomes in intensive care unit (ICU) specialists facing the COVID-19 outbreak.	85 countries	*n* = 1001 ICU staff Men: 659 Women: 342	Burnout (MBI) and anxiety and depression (HADS): Factors independently associated with symptoms of severe burnout included age (hazard ratio [HR] = 0.98/year [0.97–0.99]) and the physicians’ rating of the ethical climate (HR = 0.76 [0.69–0.82]). Factors associated with the prevalence of depression symptoms were female sex (42.4% vs. 30.5%; *p* < 0.001), younger age (44 (38–51)vs. 46 years (39–54), *p* = 0.004), being single (21.4% vs. 15.1%, *p* = 0.030), living in a city with >1 million inhabitants (63.6% vs. 49.5%, *p* < 0.001), and greater religiosity (36 (5–67) vs. 0.21 (0–59), *p* = 0.002). Factors that were independently associated with anxiety symptoms were female sex (HR = 1.85 [1.33–2.55]), work in a university hospital (HR = 0.58 (0.42–0.80)), life in a city with >1 million inhabitants (HR = 1.40 (1.01–1.94)), and the physicians’ rating of the ethical climate (HR = 0.83 (0.77–0.90)).
Barello et al. (2020a) [8]	To describe the burnout levels of a sample of Italian healthcare workers involved in the management of the COVID-19 pandemic. To explore the relationship between professional burnout and psychosomatic symptoms, and perceived work demands and work resources.	Italy *n* = 532	Nurses: 327 Physicians: 106 Others: 99 Men: 133 Women: 399	Burnout (MBI): 41% showed elevated levels of emotional exhaustion and 27% showed elevated levels of depersonalization. The perceived levels of professional risk *r* = 0.360 (*p* < 0.001), emotional demands *r* = 0.244 (*p* < 0.001), uncertainty of the clinical situation *r* = 0.284 (*p* < 0.001), and conflict between work and health professionals’ families *r* = 0.146 (*p* = 0.001), were correlated with the experience of burnout and especially with emotional exhaustion. The feeling that one’s work has meaning (*r* = −0.344, *p* < 0.001) and being inspired by work (*r* = −0.316, *p* < 0.001) was negatively correlated with emotional exhaustion (*r* = 0.316, *p* < 0.001) and depersonalization (*r* = 0.280, *p* < 0.001), and positively correlated with personal accomplishment (*r* = 0.435, *p* < 0.001).
Barello et al. (2020b) [43]	To report work-related psychological pressure, emotional exhaustion, and somatic symptoms during the COVID-19 outbreak among healthcare workers in Italy.	Italy	*n* = 376 Hospital services: 307 Rehabilitation centers: 6 Outpatient: 3 Private centers: 1 Others: 55 Not specified: 4 Men: 99 (26.3%) Women: 277 (73.7%)	Burnout (MBI): Mean emotional exhaustion score *M* = 22.7 (12.1%); depersonalization *M* = 6.1 (5.7%); personal accomplishment *M* = 37.5 (7.6%). The results showed a strong effect of gender on emotional exhaustion (*F* (1.312) = 12.444; *p* < 0.001; ηp2 = 0.038), with women showing higher levels than men (*M* = 24.05 ± 11.57 and *M* = 18.74 ± 12.65, respectively).
Blanco-Donoso et al. (2020) [44]	To analyze the psychological consequences (secondary traumatic stress and fear of contagion) of COVID-19 in workers in nursing homes for the elderly. To study the influence of stressors (workload, social pressure from work, and contact with death and suffering) and inadequate work resources (lack of personnel, materials, and PPE, and insufficient support from coworkers and supervisors).	Spain	*n* = 228 Physicians: 7.4% Nurses: 19.3% Nursing assistants: 30.3% Geriatric assistants: 6.5% Social workers: 15.3% Psychologists: 7.9% Occupational therapists: 4.8% Physiotherapists: 2.6% Managers: 1.3% Men: 19.7% Women: 80.3%	Secondary trauma (STSQ): Workers in contact with COVID-19 patients (*n* = 156) *M* = 2.80; workers not in contact with COVID-19 patients (*n* = 70) *M* = 2.62. Workload stressors were related to secondary traumatic stress (*r* = 0.40, *p* < 0.01; *r* = 0.47, *p* < 0.01; *r* = 0.45, *p* < 0.01, respectively). Only the lack of personnel and PPE were associated with secondary traumatic stress (*r* = 0.33, *p <* 0.01). Secondary trauma was related to fear of contagion (*r* = 0.38, *p <* 0.01). Professionals in contact with patients who had tested positive for COVID-19 showed higher levels of secondary traumatic stress than professionals working in nursing homes with no detected cases (*M* = 2.80 > *M* = 2.62; *t* = 3, 05, *p* < 0.01, *d* = 0.46).
Buselli et al. (2020) [45]	To identify the contribution of personal- and work-contextual variables (gender, work position, years of experience, and proximity to infected patients) in the dimensions of professional life (satisfaction, burnout, and secondary trauma). To analyze the impact of these dimensions on health workers’ mental health.	Italy	*n* = 256 Physicians: 85 Nurses: 133 Others: 47 Men: 84 Women: 181	Burnout, secondary trauma, compassion satisfaction (ProQOL-5), generalized anxiety (GAD-7), and depression (PHQ-9): Mean burnout score *M* = 19.8 ± 5.0. Mean secondary trauma score *M* = 18.0 ± 5.6. Average compassion satisfaction score *M* = 38.2 *SD* = 7.0. Burnout scores (*b* = 0.400, *p* < 0.001) and secondary traumatic stress (*b* = 0.200, *p* = 0.007) showed a significant positive association with depression scores. Frontline activity (*b* = 1.760, *p* = 0.008) and work in an ICU (*b* = 2.290, *p* = 0.001) was significantly associated with anxiety and depression scores. The mean anxiety and depression scores were *M* = 4.2 and *M* = 4.5.
Çelmeçe and Menekay (2020) [46]	To determine the effect of the levels of stress, anxiety, and exhaustion on the quality of life of health professionals who actively worked in hospitals during the COVID-19 pandemic.	Cyprus	*n* = 240 Physicians: 70 Nurses: 120 Assistant nurses: 50 Men: 70 Women: 170	Burnout (MBI) anxiety (STAI) quality of life (QLS), and stress (PSS): Considering their occupational situations, the mean burnout scores of nurses were significantly higher (*M* = 26.98 ± 8.91, *t* = 2.564, *p* = 0.004) than those of physicians (*M* = 29.34 ± 8.41) and auxiliary personnel (*M* = 24.15 ± 8.14). Burnout was significantly positively correlated with stress (*r* = 0.46, *p* < 0.001) and trait anxiety (*r* = 0.59, *p* < 0.001), and negatively with quality of life (*r* = −0.700, *p* < 0.001). Stress (*t* = −2.392, *p* = 0.017), trait anxiety (*t* = 3.214, *p* = 0.001), and quality of life (*t* = 0.73, *p* = 0.011) scores were higher in women than in men.
Chen et al. (2020a) [47]	To assess the prevalence of anxiety and depression symptoms in healthcare workers during the COVID-19 pandemic, and to identify the associated risk factors.	China	*n* = 902 Physicians: 543 Nurses: 311 Others: 48 Men: 283 Women: 619	Burnout (CMBI), generalized anxiety (GAD-7), and depression (PHQ-9): Work burnout was a risk factor for anxiety (odds ratio OR 1, 939 (1, 276–2, 988) and depression (OR 3, 121 (2, 028–4, 913)); 150 (16.63%) health workers experienced moderate–severe anxiety and 165 (18.29%) had symptoms of moderate–severe depression; 36.2% simultaneously had anxiety and depression. The rates of occurrence of moderate–severe anxiety and depression were significantly higher in those who experienced increased workloads (*p* = 0.001 and *p* < 0.001, respectively).
Chen et al. (2020b) [48]	To assess trauma, burnout, growth, and associated factors in nurses who worked during the COVID-19 pandemic.	Several countries	*n* = 12,596 ICU: 660 Pulmonary medicine: 419 Infectious diseases: 208 Emergencies: 702 Others: 10,594 Men: 555 Women: 12,041	Burnout (MBI) and post-traumatic growth (PTGI-SF): Burnout: the depersonalization score was moderate *M* = 5.5 ± 4.6. Women, critical unit workers, and staff in departments related to COVID-19 had significantly higher depersonalization scores (*p* < 0.001). The average score for lack of personal accomplishment was *M* = 19 ± 8.4; participants experienced a small lack of personal accomplishment. Post-traumatic growth: the mean post-traumatic growth score was *M* = 28.0 ± 11.5; participants scoring in the 60th percentile or higher showed personal growth because of the effects of COVID-19.
Chor et al. (2020) [49]	To assess the prevalence of burnout among physicians and nurses in the emergency services and urgent care centers of a regional health group through a cross-sectional study.	United States of America	*n* = 337 Nurses: 210 Physicians: 127 Men: 32.3% Women: 67.7%	Burnout (CBI): The mean personal exhaustion score was 49.2 ± 18.6, ranging from moderate to light (49.3%). Nurses had more burnout (*M* = 51.3 ± 19.6) than physicians (*M* = 45.7 ± 16.2; *p* = 0.005). Staff who had worked in emergencies or in an urgent care center prior to the COVID-19 pandemic also had a higher rate of moderate–severe personal burnout compared to those deployed from other departments (90.4% vs. 9.6%, *p* = 0.004).
Civantos et al. (2020a) [50]	To assess mental health symptoms among head and neck surgeons in Brazil during the COVID-19 pandemic.	Brazil	*n* = 163 head and neck surgeons Men: 121 Women: 42	Burnout (Mini-Z MBI), TEPT (IES-R), and generalized anxiety (GAD-7): Burnout was reported in 24 (14.7%) participants, and in a significantly higher number of women compared to men (*p* = 0.007); 4.9% obtained a score equivalent to probable PTSD; 74 participants (45.5%) presented anxiety symptoms, with 25.8% in a mild range, 11.7% in a moderate range, and 8.0% in a severe range. Women reported a greater increase in anxiety symptoms compared to men (*p* = 0.001).
Civantos et al. (2020b) [51]	To assess the symptoms of exhaustion, anxiety, distress, and depression in the participants.	United States of America	*n* = 349 Medical residents: 165 Physicians: 184 Men: 212 Women: 137	Burnout (Mini-Z MBI), generalized anxiety (GAD-7), and depression (PHQ-2): Burnout was reported in 76 (21.8%) participants. The burnout prevalence was higher in residents compared to doctors (49 [29.7%] vs. 27 [14.7%], *p* = 0.001). Women had a higher prevalence compared to men (40 [29.2%] vs. 36 [17.0%], *p* = 0.010). Some 210 participants (60.2%) had depression: 32.7% mild, 20.9% moderate, and 6.6% severe. Women had higher scores than men (*p* = 0.001). A total of 167 participants presented symptoms of anxiety (47.9%): 28.9% mild, 11.5% moderate, and 7.4% severe. Women had higher scores (*p* = 0.001), and 37 participants (10.6%) met the criteria for a diagnosis of depression.
Coleman et al. (2020) [52]	Gain insight into the experience of early-career surgeons and surgery residents at the American College of Surgeons.	United States of America	*n* = 1160 Surgery residents: 465 Early-career surgeons: 695 Men: 53% Women: 47%	Burnout (aMBI), anxiety, and depression (PHQ-9): 55% of the residents reported emotional exhaustion, 39% depersonalization, and 45% a decrease in the sense of personal accomplishment. Similarly, 56% of early-career surgeons reported emotional exhaustion, 30% depersonalization, and 45% a decreased sense of personal accomplishment. Some 31% of the residents reported depressed mood, 54% had anxiety, 37% experienced changes in sleep habits, 22% changes in appetite, 31% decreased interest or happiness in activities, 39% a change in weight, and 35% had difficulty maintaining attention. In turn, 31% of early-career surgeons reported symptoms of depression, 61% anxiety, 42% changes in sleep habits, 21% changes in appetite, 36% a lack of interest, 44% a change in weight, and 34% a decrease in the maintenance of care.
Correia and Almeida (2020) [53]	To identify the main psychosocial variables that could have been protective factors for burnout in physicians and nurses in the first 2 months of the COVID-19 pandemic in Portugal.	Portugal	*n* = 497 Physicians: 229 (48% men and 52% women) Nurses: 268 (26.9% men and 73.1% women)	Burnout (OLBI), empathy (BES), and work areas (AWS): Emotional exhaustion: physicians (*M* = 3.07 ± 0.65); nurses (*M* = 3.10 ± 0.60). Depersonalization: physicians (*M* = 2.69 ± 0.71); nurses (*M* = 2.76 ± 0.70). Physicians: there were statistically significant negative correlations between burnout and depersonalization, and income; confidence in COVID-19 policies; peer justice; distributive justice; procedural justice; and professional identification. There was a positive significant association between workload, and burnout and disengagement. Cognitive empathy, meaningful work, patient fairness, and patient family fairness were negatively correlated with disconnection. Male sex was also a risk factor for disconnection. Nurses: significant negative correlations for burnout and depersonalization, and income; confidence in COVID-19 policies; meaningful work; peer justice; patient justice; family patient justice; distributive justice; procedural justice; and professional identification. Burnout was positively associated with affective empathy and negatively associated with age, years of professional experience, and COVID-19 task changes. Female sex was a risk factor for burnout. Disconnection was significantly negatively related to religion. Burnout and disconnection were positively associated with workload.
Cravero et al. (2020) [54]	To determine how the level of exposure to COVID-19 patients affected the perceived safety, training, and well-being of residents and fellows.	China, Saudi Arabia, and Taiwan	*n* = 1420 Medical residents: 1101 Physicians: 319 Medical residents exposed to > 60 COVID-19 patients: 111 Physicians exposed to > 60 COVID-19 patients: 14 Men: 51% Women: 49%	Burnout (MBI): 66% of the residents who cared for more than 60 COVID-19 patients had burnout, while 39% of the residents who did not see COVID-19 patients reported burnout (*p* < 0.001). The number of patients attended to was a predictor of burnout: attending to > 60 COVID-19 patients (adjusted odds ratio [AOR] = 4.03; 95% confidence interval [CI; 2.12–7.63]). Access to adequate PPE ‘most of the time’ (AOR = 1.99; 95% CI [1.41–2.80]) or ‘sometimes’ (AOR = 2.81; 95% CI [1.60–4.91]) was significantly associated with exhaustion.
Demartini et al. (2020) [55]	To assess the psychopathological impact of the pandemic on the general population of Lombardy and compare the prevalence of psychiatric symptoms among the general population and healthcare workers there.	Italy	*n* = 432 General population: 309 Health workers: 123 Men: 28% Women: 72%	Burnout (MBI), anxiety and depression (DASS-21), and insomnia (PQSI): 38% of health workers had symptoms of emotional exhaustion, 39.8% depersonalization, and 48% personal accomplishment. In the general population, 9.06% had symptoms of emotional exhaustion, 49% depersonalization, and 42.9% high personal accomplishment; A total of 59 health workers presented pathological levels of stress (48%), 47 anxiety (38.2%), 51 depression (41.5%), and 88 ‘poor sleep’ (71.5%); 28 participants from the general population presented pathological levels of stress (57.1%), 23 anxiety (46.9%), 25 depression (51%), and 35 ‘poor sleep’ (71.4%).
De Wit et al. (2020) [56]	To report on burnout and describe the psychological effects of working as a Canadian emergency physician during the first weeks of the COVID-19 pandemic.	Canada	*n* = 468 Emergency residents: 40 (9%) Physicians: 428 (91%) Men: 240 (51%) Women: 227 (49%)	Burnout (MBI): Levels did not change significantly in the period considered (emotional exhaustion *p* = 0.632; depersonalization *p* = 0.155). Getting tested for COVID-19 and the number of shifts worked was associated with high emotional exhaustion and high depersonalization levels (OR 1.3; 95% CI [1.1–1.5] and OR 4.3; 95% CI [1.1–17.8], respectively). Well-being was influenced by PPE, safety measures, the number of patients in their care, and training on the disease (data not specified).
Di Monte et al. (2020) [57]	To explore the relationships between the dimensions of exhaustion and various psychological characteristics in Italian primary care physicians during the COVID-19 emergency.	Italy	*n* = 102 primary care physicians Men: 38 Women: 64	Burnout (MBI), coping with stressful situations (CISS), and resilience (RS-14): The coping style for stress emotions positively predicted emotional exhaustion (β = 0.461, *p* < 0.001). Task-oriented and emotion-oriented coping were significant predictors of depersonalization (respectively, β = 0.183, *p* = 0.034; β = −0.298, *p* = 0.023). Resilience positively predicted personal achievement (β = 0.500; *p* < 0.001). People with elevated levels of burnout showed less resilience and more often adopted a task-oriented coping strategy than the medium-risk group.
Dimitriu et al. (2020) [58]	To measure the prevalence of burnout syndrome during the COVID-19 pandemic in resident physicians.	Romania	*n* = 100 Group A Emergency residents: 30 ICU residents: 10 Radiology residents: 10 Group B Gynecology residents:15 Orthopedic residents: 10 General surgery residents: 25	Burnout (MBI): The overall prevalence of burnout syndrome among resident physicians was high. Burnout was significantly more common in residents who worked in regular services (group B) compared to resident physicians who worked in services considered frontline departments (group A; 86% vs. 66%, *p* = 0.050).
Dincer and Inangil (2020) [59]	To investigate the effectiveness of implementation of the emotional freedom technique (EFT) program in the prevention of stress, anxiety, and burnout in nurses treating COVID-19.	Turkey	*n* = 80 Nurses Men: 8 Women: 64	Burnout (Burnout Scale), Subjective relief services (SUD), and anxiousness (STAI): Burnout, pre-intervention: EFT group *M* = 3.62 ± 0.76; control *M* = 3.56 ± 0.72. Post-intervention: EFT group *M* = 2.48 ± 1.06; control *M* = 3.43 ± 0.76. Subjective relief services, pre-intervention: EFT group *M* = 7.82 ± 1.33; control: *M* = 7.48 ± 1.36. Post-intervention: EFT group *M* = 2.85 ± 1.21; control *M* = 7.40 ± 1.53. Anxiety, pre-intervention: EFT group *M* = 67.68 ± 9.05; control *M* = 64.7 ± = 8.05. Post-intervention: EFT group *M* = 32.25 ± 4.67; control *M* = 64.43 ± 7.68. Significant differences were observed between the groups in the three variables studied after the intervention and between the pre-test and post-test in the EFT group (*p* < 0.001).
Dinibutun (2020) [60]	To assess the prevalence and extent of burnout among physicians and to investigate factors related to burnout and the influence of COVID-19 on burnout syndrome.	Turkey	*n* = 200 physicians Men: 41% Women: 59%	Burnout (MBI): emotional exhaustion: *M* = 3.00 ± 0.62, average levels; depersonalization: *M* = 2.00 ± 1.01), low levels; self-fulfillment: *M* = 2.00 ± 0.57), low levels; total burnout: *M* = 2.50 ± 0.43, low levels.
Dobson (2020) [61]	To examine psychological distress in healthcare workers during the COVID-19 pandemic in April–May 2020.	Australia	*n* = 320 Senior medical staff: 58 Junior medical staff: 41 Nurses: 86 Others: 131 Men: 58 (18.4) Women: 248 (78.5)	Burnout (PFI), TEPT (IES-R), depression (PHQ-9) resilience (CD-RISC10), and generalized anxiety (GAD-7): 83 participants (29.5%) experienced burnout symptoms. The rates of burnout, depression, anxiety, and PTSD differed between professionals; senior medical staff reported the lowest levels of psychological distress. Frontline workers reported elevated levels of resilience and less severe psychological distress compared to other healthcare workers. Work in a high-exposure environment was associated with greater PTSD symptoms (*t* (279)= 2.26, *p* = 0.024) and burnout (*t* (270) = 2.03, *p* = 0.044). A subset of participants experienced moderate–severe symptoms of depression (21%), anxiety (20%), and PTSD (29%); 23 participants (8.1%) felt suicidal ideation.
Dosil et al. (2020) [62]	To measure the levels of stress, anxiety, depression, compassion fatigue, and post-traumatic stress symptoms among healthcare professionals in Spain after flattening of the COVID-19 infection rate curve. To study the possible differences in these symptoms according to other relevant factors such as age, sex, professional category, contact with COVID-19, and perception of social compliance with health measures.	Spain	*n* = 973 Physicians: 433 (44.5%) Nurses: 318 (32.6%) Technicians/assistants: 222 (22.9%) Men:165 (16.5%) Women: 808 (82.9%)	Burnout, secondary traumatic stress, and compassion satisfaction (ProQOL-5); depression and anxiety (DASS-21); and post-traumatic stress (PCL-C): Burnout levels were generally medium (90.6%); the oldest participants in the sample (36–55 and < 56) showed more burnout than the younger participants. Burnout was higher in physicians than in nurses and no difference was found between nurses and technicians/assistants. There were no differences in burnout between those who had or had not been in direct contact with COVID-19. Secondary traumatic stress levels were low (0.2% high and 19.2% medium). Compassion satisfaction levels were high (33.2%) or medium (63.1%). The highest levels of compassion satisfaction were found among participants aged 26–35 and 36–55 years. Compassion satisfaction was higher in technicians/assistants than in nurses, while it was higher in nurses than in physicians. Levels of depression, anxiety, stress, and post-traumatic stress were significantly higher in nurses and technicians/assistants than in physicians.
Duarte et al. (2020) [63]	To assess the relationship between sociodemographic and mental health variables in health workers in terms of three dimensions of exhaustion: personal, work-related, and client-related.	Portugal	*n* = 2008 Healthcare technicians: 707 (35.2%) Physicians: 511 (25.4%) Nurses: 409 (20.4%) Pharmacists: 88 (4.4%) Psychologists: 83 (4.1%) Nutritionists: 72 (3.6%) Men: 330 (16.4%) Women: 1678 (83.6%)	Burnout (CBI): 1066 participants (more than 50%) showed high levels of work-related burnout. Potential risk predictors for burnout were gender, parental status, marital status, and salary reduction. A higher level of exhaustion was found in women compared to men (*p* < 0.001). Frontline workers showed higher levels of personal, work-related, and patient-related burnout (β = 4.24, β = 3.91, and β = 2.35, *p* < 0.001, respectively). Workers in direct contact with COVID-19 patients presented higher levels of personal (β = 3.27, *p* < 0.001) and work-related exhaustion (β = 3.45, *p* < 0.001).
Elhadi et al. (2020) [64]	To determine the prevalence of burnout among hospital healthcare workers in Libya during the COVID-19 pandemic and the ongoing civil war.	Libya	*n* = 532 Internal medicine: 223 (41.9%) Intensive care: 64 (12%) Emergencies: 111 (20.9%) Surgery: 134 (25.2%) Men: 294 (55.3%) Women: 238 (44.7%)	Burnout (aMBI): Emotional exhaustion: 357 (67.1%) participants reported high levels (≥ 10); the average score was 11.3 ± 4.8. A statistically significant association was found between emotional exhaustion and gender, years of work experience, service, and life in a conflict area (*p* < 0.050). Depersonalization: 252 (47.4%) participants showed elevated levels (≥ 10); the average score was 8.5 ± 5.1. The influencing characteristics were sex, age, department, internal displacement, and verbal abuse (*p* < 0.050). Personal accomplishment: 121 (22.7%) participants showed a lower sense of personal achievement (≤10); the average score was 12.7 ± 3.7, but did not significantly correlate with other variables.
El Haj et al. (2020) [65]	To measure the level of burnout in health workers in a geriatric nursing home during the COVID-19 crisis.	France	*n* = 84 Nursing assistants: 33 Nurses: 25 Physicians: 15 Others: 11 Men: 25 Women: 59	Burnout (OLBI): Nursing assistants: *M* = 37.66 ± 3.32; nurses: *M* = 38.89 ± 3.32; physicians: *M* = 37.21 ± 3.19; others: *M* = 39.42 ± 3.76. No significant differences were found between professional categories: χ^2^ (1, *n* = 84) = 0.36, *p* = 0.550.
Franza et al. (2020a) [66]	To investigate stress, burnout, and compassion fatigue in healthcare workers.	Italy	*n* = 102 Resident physicians: 12 (11.76%) Psychologists: 5 (4.90%) Nurses: 24 (23.52%) Therapists (psychiatry, respiratory therapy, physical, occupational, and speech): 21 (20.59%) Clinical social workers: 30 (29.41%) Technicians: 4 (3.92%) Administrative: 6 (5.88%) Men: 48 Women: 54	Burnout (CBI), secondary trauma (STSS), and compassion fatigue (FCs): The overall compassion fatigue scores were increased in all the workers; the largest increase was in psychiatric and multidisciplinary health workers (22% and 33%, respectively). The job category with the highest percentage of burnout (39.67% vs. 40.67%) was social workers in the psychiatric and multidisciplinary departments. Compassion fatigue was reduced among therapists and nurses, and the scores were also lower on the burnout and secondary trauma subscales compared to physicians and psychologists.
Franza, et al. (2020b) [67]	To investigate stress, burnout, and compassion fatigue in healthcare workers.	Italy	*n* = 102 Resident physicians: 12 (11.76%) Psychologists: 5 (4.90%) Nurses: 24 (23.52%) Therapists (psychiatry, respiratory, physical, occupational, and speech): 21 (20.59%) Clinical social workers: 30 (29.41%) Technicians: 4 (3.92%) Administrative: 6 (5.88%) Men: 48 Women: 54	Burnout (CBI), compassion fatigue (FCs), professional quality of life (ProQOL-5), and hopelessness (BHS): Nurses showed a high percentage of compassion fatigue and the lowest average scores on the ProQOL subscale of vicarious traumatic stress (45.83% with moderate to severe scores). There was a greater increase in the mean values of burnout in all the groups analyzed (*p* = 0.003). About half of the respondents scored above average (37.5%) on the hopelessness scale.
Giusti et al. (2020) [68]	To identify the prevalence of burnout and psychological suffering in health professionals during the early phases of the COVID-19 pandemic.	Italy	*n* = 330 Physicians: 140 Nurses: 86 Nursing assistants: 38 Psychologists: 35 Others: 32 Men: 124 Women: 206	Burnout (MBI), anxiety and depression (DASS-21), and TEPT (IES-6): Emotional exhaustion: *M* = 22.3 ± 11.4; 107 (35.7%) had moderate scores, and 105 (31.9%) had severe levels of emotional exhaustion. Anxiety: *M* = 3.3 ± 3.6. there were clinical levels of anxiety in 103 (31.3%) participants. Depression: *M* = 4.0 ± 4.2. clinical levels of depression were identified in 88 participants (26.8%). Post-traumatic stress: *M* = 3.2 ± 2.1; 36.7% of the participants reported post-traumatic stress.
Hoseinabadi et al. (2020) [69]	To assess the level of burnout during the COVID-19 outbreak and identify the influencing factors in frontline nurses and non-frontline nurses.	Iran	*n* = 245 nurses COVID-19 exposure: Men: 82 (54.3%) Women: 69 (45.7%) No COVID-19 exposure: Men: 45 (52.1%) Women: 49 (47.9%)	Burnout (OLBI): The occupational stress (*p* = 0.006) and burnout (*p* = 0.002) scores in the exposure group were significantly higher than in the non-exposure group. Work stress was the only factor that was significantly linked to COVID-19-related burnout (β = 0.308, *p* = 0.031).
Hu et al. (2020) [70]	To assess mental health (burnout, anxiety, depression, and fear) and its associated factors among frontline nurses caring for COVID-19 patients in Wuhan, China.	China	*n* = 2014 nurses Men: 260 (12.9%) Women: 1754 (87.1%)	Burnout (MBI), anxiety (SAS), depression (SDS), self-efficacy (GSS), resilience (CD-RISC-10), social support (MSPSS), and fear (FS-HPs): Emotional exhaustion (*M* = 23.44 ± 13.80) correlated positively with skin lesions (*r* = 0.182) and negatively with self-efficacy (*r* = 0.193), resilience (*r* = 0.325), intra-family social support (*r* = 0.170), and extra-family social support (*r* = 0.234). Depersonalization (*M* = 6.77 ± 7.05) was negatively correlated with resilience (*r* = 0.208), intra-family social support (*r* = 0.221), and extra-family social support (*r* = 0.216). Personal accomplishment (*M* = 34.83 ± 9.95) was positively correlated with self-efficacy (*r* = 0.376), resilience (*r* = 0.436), intra-family social support (*r* = 0.348), and extra-family social support (*r* = 0.363); 67 obtained anxiety scores (3.3%), and 23 obtained severe depression scores (1.1%). Fear was negatively correlated with resilience (*r* = 0.121).
Jose et al. (2020) [71]	To assess the burnout and resilience of frontline emergency nurses in a tertiary care facility.	India	*n* = 120 nurses Men: 32 (26.7%) Women: 88 (73.3%)	Burnout (MBI-HSS) and resilience (CD-RISC): Burnout: more than half of the nurses (54%) reported a high level of emotional exhaustion and 37% reported a moderate level. Approximately 52% of the participants expressed a moderate level of depersonalization; 78.5% experienced average levels of personal accomplishment. Resilience: 47.5% of the frontline nurses expressed a moderate–high level of resilience, 53.3% a moderate level of self-efficacy, and 45.8% a moderate level of optimism.
Kannampallil et al. (2020) [72]	Investigate the effects of the exposure of fellows and residents to COVID-19 patients on depression, anxiety, stress, burnout, and job satisfaction.	United States of America	*n* = 393 Fellows: 132 Residents: 261 Exposed to COVID-19: 218 Not exposed to COVID-19: 179 Men: 175 Women: 218	Burnout (PFI), and depression and anxiety (DASS-21): The group of health workers exposed to COVID-19 had higher scores for burnout compared to those not exposed (46.3% and 33.7%, *p* = 0.011). The exposed group had a higher prevalence of stress (29.4% and 18.9%, *p* = 0.016); there were no differences in anxiety (21.6% and 14.9%, *p* = 0.089) and both groups had similar rates of depression (28% and 26.3%, *p* = 0.700). Participants who were exposed to patients with COVID-19 reported significantly greater levels of stress compared to those who were not exposed to patients with COVID-19 (10.96; 95% CI [9.65–12.46] vs. 8.44; 95% CI [7.3–9.76]; *p* = 0.043).
Kelker et al. (2020) [73]	To assess the well-being, resilience, burnout, and well-being factors of emergency physicians and residents during the early phase of the COVID-19 pandemic.	United States of America	*n* = 213 Emergency physicians: 157 (74%) Residents: 56 (26%) Men: 46% Women: 54%	Burnout (PWLS), well-being (WBI), and resilience (BRS): The following dimensions were evaluated over 4 weeks: exhaustion did not significantly change (30% to 22%; *p* = 0.390); working part-time had twice the risk of burnout (OR 2.45; 95% CI [1.10–5.47]); well-being improved (30% to 14%; *p* = 0.010); and symptoms of stress, anxiety, or fear were initially 83% and decreased to 66% (*p* = 0.009). The initial resilience levels were normal–high.
Khalafallah et al. (2020a) [74]	Investigate the impact of the COVID-19 pandemic on the workflow, burnout, and career satisfaction of neurosurgery residents in the US.	United States of America	*n* = 111 neurosurgery residents Men: 73 Women: 37	Burnout (aMBI): low levels of emotional exhaustion (51.4%); low levels of depersonalization (67.6%); and elevated levels of personal achievement were noted (78.4%).
Khalafallah et al. (2020b) [75]	To investigate the impact of the pandemic on burnout and job satisfaction among neurosurgeons in the US.	United States of America	*n* = 407 neurosurgeons Neurosurgery subspecialties: None: 150 (36.9%) Spinal column/peripheral nerves: 87 (21.4%) Cerebrovascular: 55 (13.5%) Pediatrics: 53 (13.0%) Neuro–Oncology: 37 (9.1%) Functional/stereotactic: 35 (8.6%) Endovascular: 26 (6.4%) Critical care: 15 (3.7%) Men: 361 (88.7%) Women: 46 (11.3%)	Burnout (aMBI): Most of the respondents reported low levels of emotional exhaustion (51.6%), low levels of depersonalization (87.5%), and high levels of personal accomplishment (81.1%). Neurosurgeons satisfied with their careers were less likely to have received subspecialty training in the spine/peripheral nerves (*p* = 0.028) and were less likely to feel that their professional life had worsened because of COVID-19 (*p* = 0.045).
Khasne et al. (2020) [76]	To study the prevalence of burnout due to the COVID-19 pandemic in India.	India	*n* = 1117 Physicians: 1667 Nurses: 198 Administration staff: 90 Paramedics (dietitians, physiotherapists, pharmacists, etc.): 43 Others: 28 Men: 55% Women: 45%	Burnout (CBI): The mean scores were as follows: personal exhaustion: *M* = 49.72 ± 18.68; work exhaustion: *M* = 39.69 ± 20.43; and exhaustion related to the pandemic: *M* = 51.37 ± 15.12. The prevalence of personal exhaustion (41.3% vs. 48.6%) and work-related exhaustion (25.0% vs. 29.1%) was significantly higher (*p* < 0.010) among surveyed women; compared to men, the odds ratio of experiencing personal- and work-exhaustion were 1.35, 95% CI [1.13–1.61] (*p* < 0.010), and 1.24, 95% CI [1.01–1.50] (*p* < 0.030), respectively.
Khattak et al. (2020) [77]	To examine the impact of the fear of COVID-19 on nurses’ intention to change their shift rotation, secondary trauma, and psychological distress.	Pakistan	*n* = 380 nurses Men: 60 Women: 320	Secondary trauma, fear, intention to change shift rotations, and leadership support (ad-hoc): The results of the regression analysis showed that fear of COVID-19 had had a positive and significant effect on secondary trauma (*b* = 4.84, *p* < 0.050), psychological distress (*b* = 4.83, *p* < 0.050), and nurses’ intention to change their shift rotation (*b* = 4.79, *p* < 0.050). Furthermore, secondary trauma and the intention to change the shift rotation were lower when leadership support was high.
Lange et al. (2020) [78]	To assess the psychological impact of COVID-19 on French community pharmacists.	France	*n* = 135 pharmacists Men: 57 Women: 78	Burnout (MBI) and TEPT (IES-R): The mean scores were PTSD = 20.6 ± 15.1; emotional exhaustion = 23.0 ± 11.4; depersonalization = 10.9 ± 5.5; and personal accomplishment = 48.1 ± 7.2. A total of 23 pharmacists (17%) reported post-traumatic stress; 33 (25%) emotional exhaustion; 46 (34.9%) depersonalization; and 4 (3%) low personal accomplishment. Women scored higher than men for post-traumatic stress disorder (*p* < 0.010) and depersonalization (*p* < 0.001).
Lázaro-Pérez et al. (2020) [78]	To find out if health professionals have suffered anxiety in relation to the death processes of their patients, and what variables were involved in this sense.	Spain	*n* = 157 Physicians: 22 (14.0%) Nurses: 109 (69.4%) Others: 26 (16.6%) Men: 33 (21.0%) Women: 124 (79.0%)	Burnout (MBI) and anxiety towards death (patients’ anxiety about death): 58.6% showed low levels of emotional exhaustion and 41.4% medium–high levels; 31.8% showed low levels of depersonalization, while 68.2% presented medium–high levels; 45.9% showed low levels of personal accomplishment; 54.1% showed medium–high levels. The risk of suffering anxiety about the death processes of patients increased by 3 points in the presence of moderate–high levels of emotional exhaustion and depersonalization.
Lee et al. (2020) [79]	To validate the effectiveness of implementing Asimov’s coping strategy to reduce emotional exhaustion in a group of primary care physicians.	Kazakhstan	*n* = 102 Physicians Intervention group (IG): 53 Control group (CG): 49 Men: 24 Women: 78	Burnout (MBI-HSS): Both groups showed elevated levels of emotional exhaustion (*p* > 0.050). After 6 months, the indicators of emotional exhaustion (*p* = 0.019) and depersonalization (*p* = 0.028) in the IG were reduced compared to the CG; this was not the case for personal accomplishment (*p* = 0.067). After 12 months, the indicators of emotional exhaustion and depersonalization decreased and personal accomplishment increased in the IG, compared to the CG (*p* < 0.050).
Li et al. (2020) [80]	To study the level of vicarious trauma in the general population, frontline nurses, and nurses who were not on the frontline during the COVID-19 pandemic.	China	*n* = 740 General population: 214 Nurses: 526 Frontline nurses: 234 Non-frontline nurses: 292 Men: 162 Women: 578	Vicarious trauma (VTS): Vicarious trauma scores for frontline nurses, including scores for physiological and psychological responses, were significantly lower than non-frontline nurses (*p* < 0.001) and the general population (*p* < 0.001). No differences were observed between the general population and non-frontline nurses (*p* > 0.050).
Litam and Balkin (2020) [81]	To investigate how moral damage affected health workers during the COVID-19 pandemic.	United States of America	*n* = 119 Physicians: 40 Nurses: 62 Other professions: 7 Men: 26 Women: 83	Secondary traumatic stress (ProQOL-5) and moral damage (MIES): Secondary traumatic stress was significantly associated with moral damage, representing only 8.4% of the variance in the model, and with a negative relationship (*r* = −0.49) with respect to moral damage. The increase in secondary traumatic stress was associated with a stronger probability of suffering moral injury.
Liu et al. (2020) [7]	To find out factors related to job burnout in Chinese health workers.	China	*n* = 830 Physicians: 564 Nurses: 316 Men: 279 Women: 601	Burnout (CMBI): A total of 80 (9.09%) respondents showed emotional exhaustion, 445 (50.57%) depersonalization, and 498 (56.59%) had reduced personal accomplishment. There were no statistical differences between the three dimensions according to gender or occupational groups.
Luceño-Moreno et al. (2020) [82]	To assess the symptoms of post-traumatic stress, anxiety, depression, and levels of burnout and resilience in Spanish health workers during the COVID-19 pandemic, as well as the relationship between these factors.	Spain	*n* = 1422 health workers Men: 194 Women: 1228	Burnout (MBI-HHS), TEPT (IES-R), anxiety and depression (HADS), and resilience (BRS): Emotional exhaustion: 584 (41%) showed high scores; there were significant correlations (*p* < 0.050) with intrusion (*r* = 0.374), avoidance (*r* = 0.345), hyperarousal (*r* = 0.423), post-traumatic stress (*r* = 0.420), anxiety (*r* = 0.512), and depression (*r* = 0.484). Depersonalization: 216 (15.2%) showed high scores; there were differences between men and women and significant correlations (*p* < 0.050) with intrusion (*r* = 0.171), avoidance (*r* = 0.219), hyperarousal (*r* = 0.218), post-traumatic stress (*r* = 0.225), anxiety (*r* = 0.289), and depression (*r* = 0.294). Personal accomplishment: 1164 (81.9%) showed high scores; there were significant correlations (*p* < 0.050) with anxiety (*r* = −0.160), and depression (r = −0.298). Anxiety: 295 (20.7%) showed severe levels of disorder. Depression: 82 (5.3%) showed severe disorder; post-traumatic stress: 805 (56.6%) showed levels of psychiatric disorder. There were gender differences in the symptoms of PTSD, anxiety, and depression. The levels of resilience were moderate: 3.02 ± 0.39.
Manzano-García and Ayala-Calvo (2020) [83]	To study whether the perception of threat generated by the COVID-19 pandemic explained burnout in nurses, and its moderating effect on the influence of resources and demands on burnout.	Spain	*n* = 771 nurses Men: 77 Women: 694	Burnout (CESQT) and psychosocial demand factors (UNIPSICO): Burnout: *M* = 42.39 ± 11.39. Autonomy (*r* = −0.227, *p* < 0.010), social support (*r* = −0.508, *p* < 0.010), and material and human resources (*r* = −0.404, *p* < 0.010) were negatively correlated with burnout; role conflicts (*r* = 0.426, *p* < 0.010), role ambiguity (*r* = 0.244, *p* < 0.010), and work overload (*r* = 0.583, *p* < 0.010), were positively correlated. The perceived threat of COVID-19 was positively correlated with exhaustion (*r* = 0.68; *p* < 0.010) and was highest for burnout and the variables used to explain it.
Martínez-López et al. (2020) [84]	To learn how the health crisis affected health professionals during the most critical weeks of the spread of the SARS-CoV-2 virus.	Spain	*n* = 157 Physicians: 22 (14.0%) Nurses: 80 (51.0%) Nursing assistants: 29 (18.5%) Others: 26 (16.5%) Men: 33 (21.0%) Women: 124 (79.0%)	Burnout (MBI): Emotional exhaustion: low, 58.6%; medium, 21.0%; high, 20.4%. Depersonalization: low, 31.8%; medium, 29.3%; high, 38.9%. Personal accomplishment: low, 45.9%; medium, 34.4%; high, 19.7%. Need for support: yes, 26.8%; no, 73.2%. Material absence of protection increased stress/anxiety: yes, 85.4%; no, 14.6%.
Matsuo et al. (2020) [85]	To assess the prevalence of burnout among frontline workers during the COVID-19 pandemic in Japan based on job categories and other factors.	Japan	*n* = 312 Nurses:126 Radiologists: 22 Pharmacists: 19 Laboratory staff: 145 Men: 89 Women: 223	Burnout (MBI): the overall prevalence of burnout was 31.4% (98 of 312 participants), and was higher in women (79 [80.6%] vs. 144 [67%], *p* = 0.020). The prevalence of burnout was significantly higher for nurses (OR 4.9; 95% CI [2.2–11.2], *p* = 0.001), laboratory personnel (OR 6.1; 95% CI [2.0–18.5], *p* = 0.002), radiologists (OR 16.4; 95% CI [4.3–61.6], *p* = 0.001), and pharmacists (OR 4.9; 95% CI [1.3–19.2], *p* = 0.020). The prevalence of burnout increased in those with less experience (OR 0.93; 95% CI [0.89–0.97], *p* = 0.001), and with greater anxiety due to a lack of familiarity with PPE (OR 2.8; 95% CI [1.4–5.5], *p* = 0.002).
Miller et al. (2020) [86]	To determine the exhaustion and resilience resources available in respiratory care services.	United States of America	*n* = 221 health workers in respiratory care services	Burnout: In response to the single indicator, “Have you personally experienced burnout?”, 72.4% reported having experienced burnout in the past; 32.6% had experienced burnout in the 6 months prior; 32.5% did not use the resources available to them for exhaustion; 11.3% took time off; 11.3% performed exercises; 8.1% meditated or practiced mindfulness; 8.8% sought counseling/therapy or used personal coping strategies; and 4.4% changed their job.
Murat et al. (2020) [87]	To determine levels of stress, depression, and burnout among frontline nurses.	Turkey	*n* = 705 nurses Men: 148 (21.0%) Women: 557 (79.0%)	Burnout (MBI), depression (BDI), and perceived stress (PSS): Mean scores for burnout: emotional exhaustion = 18.9 ± 8.5; depersonalization = 7.3 ± 4.5; and personal accomplishment = 11.4 ± 5.0. Mean score for perceived stress = 31.4 ± 8.7, and for depression = 16.0 ± 9.4. More stress was perceived in public hospitals than in private ones (*M* = 35.5 ± 7.7, *M* = 33.1 ± 7.4, *t* = 14.74, *p* < 0.001). Nurses who were dissatisfied with the care they had rendered had higher perceived stress compared to those who felt competent (*M* = 33.5 ± 9.9, *M* = 30.9 ± 8.7, *t* = 7.131, *p* = 0.028). Nurses with a higher education degree who tested positive for COVID-19 showed more symptoms of depression (*p* < 0.050).
Ng et al. (2020) [88]	To assess the prevalence of burnout in oncological health professionals during the COVID-19 pandemic.	Singapore	*n* = 421 Physicians/nurses: 240 Others: 176 Missing data: 5 Men: 97 Women: 311 Others: 13	Burnout (MBI) and anxiety (GAD7): The prevalence of burnout was 43.5%, and of anxiety was 14.0%. Health workers who were younger (OR 1.83; 95% CI [1.09–3.12], *p* = 0.024), more anxious (OR 5.92; 95% CI [3.06–12.18], *p* not specified), or more fearful (OR 1.89; 95% CI [1.23–2.93], *p* = 0.004) were more likely to experience exhaustion. A perceived lack of support, public condemnation, a perceived substantial risk of contracting COVID-19, and low confidence in health facility readiness were associated with higher rates of burnout.
Osama et al. (2020) [89]	To measure the positive and negative impacts of the pandemic on surgical residency programs and on the lives of surgical residents.	Pakistan	*n* = 112 General surgery: 48 (42.8%) Neurosurgery: 18 (16.0%) Orthopedic surgery: 12 (10.7%) Plastic surgery: 10 (8.9%) Cardiothoracic surgery: 4 (3.6%) Other surgery: 6 (5.4%) Urology: 14 (12.5%) Men: 67 (59.8%) Women: 45 (40.2%)	Burnout (MBI): Emotional exhaustion decreased after the pandemic (during the peak: *M* = 6.31 ± 1.62; after: *M* = 3.77 ± 1.08; *p* = 0.008). Depersonalization decreased after the pandemic (during the peak: *M* = 3.10 ± 1.06; after: *M* = 1.00 ± 0.80; *p* < 0.001). Personal accomplishment decreased after the pandemic (during the peak: *M* = 5.33 ± 1.44; after: *M* = 3.56 ± 1.21; *p* = 0.002). Of the total number of residents, 97 (86.6%) stated that their surgical practice duration had been negatively affected by the pandemic. A total of 92 (82.1%) had their clinical exposure affected; 69 (61%) were worried about transmitting the disease to their relatives; and 43 (38.4%) said they were afraid of dying because of their direct exposure to the virus. The average number of work hours per week for surgical residents decreased (before: *M* = 81.10 ± 6.21; after: *M* = 49.16 ± 6.25; *p* < 0.001) because of the COVID-19 outbreak.
Park et al. (2020) [90]	To investigate psychological distress in infectious disease physicians during the COVID-19 disease outbreak in the Republic of Korea.	Republic of Korea	*n* = 115 physicians Men: 48 Women: 67	Burnout (MBI-HSS), depression, anxiety, and stress (DASS-21): 90.4% of the respondents met the diagnostic criteria for burnout; 20 (17.4%) met the criteria for depression, 23 (20.0%) for anxiety, and 5 (4.3%) for stress. Women had higher scores for burnout, depression, and anxiety than men.
Pinho et al. (2020) [91]	To determine the prevalence of depression, anxiety, insomnia, distress, and compassion fatigue, as well as factors related to the presence of symptoms.	Paraguay	*n* = 126 Physicians: 44 (34.9%) Nurses: 29 (23.0%) Others: 53 (42.1%) Men: 22 (17.5%) Women: 104 (82.5%)	Compassion fatigue and satisfaction (ProQOL-5); depression and anxiety (GAD-7); insomnia (ISI); and distress (IES-R): Compassion fatigue: average level *n* = 8 (14.3%); mild level *n* = 27 (21.4%); moderate level *n* = 27 (21.4%); severe level *n* = 54 (42.9%). Women (*p* = 0.048), nurses (*p* = 0.004), and physicians (*p* = 0.022), all had greater levels of compassion fatigue. The symptoms of depression (*p* = 0.023), anxiety (*p* = 0.035), insomnia (*p* = 0.024), and distress (*p* = 0.001) were more severe in women. More insomnia symptoms were presented in participants who had not attended cases of respiratory infections (*p* = 0.014). Anxiety was significantly higher in nurses compared to other health professionals (*p* = 0.038). Age and depression (*rS* = 0.253, *p* = 0.004), anxiety (*rS* = 0.228, *p* = 0.010), and distress (*rS* = 0.175, *p* = 0.0497) were significantly negatively correlated.
Prasad et al. (2020) [92]	To assess the mental health outcomes of health workers working during the COVID-19 pandemic.	United States of America	*n* = 347 Nurses: 248 Administrative staff: 63 Nursing assistants: 36 Men: 32 Women: 315	Burnout (MBI), anxiety (GAD-7) and depression (PHQ-2): Burnout: 30.0% reported burnout. Anxiety: 23.1% experienced severe distress; 69.5% experienced some type of anxiety; 68.6% indicated that their anxiety symptoms had made their work or daily routine at least “somewhat difficult” to maintain. Depression: 22.8% had symptoms of depression.
Rapisarda et al. (2020) [93]	To investigate the early impact of the COVID-19 emergency and quarantine on the well-being, working conditions, and working practices of mental health staff and professionals in Lombardy and compare the findings with the available data on health workers facing the COVID-19 outbreak.	Italy	*n* = 241 Psychologists: 73 (30.3%) Counselors: 68 (28.2%) Physicians: 28 (11.6%) Social workers: 15 (6.2%) Nurses: 27 (11.2%) Support workers: 7 (2.9%) Managers/coordinators: 7 (2.9%) Others: 11 (6.6%) Men: 56 (23.2) Women: 185 (76.8%)	Burnout (MBI), anxiety (GAD7), and depression (PHQ 9): Burnout: *M* = 16.7 ± 11.5. Outpatient services workers reported a decrease in workload (68.2%), while for inpatient services workers, the workload had remained the same (37.9%) or had increased (41.4%). Anxiety: *M* = 5.1 ± 3.4; 11.6% had scores higher than moderate anxiety. Depression: *M* = 4.7 ± 2.9; 6.6% had scores higher than moderate depression. The factors associated with burnout, anxiety, and depression were being a doctor or a woman, working in close contact with COVID-19-infected users, working in outpatient services, and the perception of a medium or high risk of contracting COVID-19 at work.
Rodriguez et al. (2020) [94]	To assess the levels of anxiety and exhaustion, changes in life, changes at home, and measures to relieve the stress of emergency physicians in the United States during the COVID-19 pandemic.	United States of America	*n* = 426 Level of medical training: Faculty: 236 Fellow: 19 Resident: 168 Men: 235 Women: 192	Burnout (ad-hoc): The emotional exhaustion of emergency physicians increased during the pandemic: median of 3 before the pandemic (IQR = 2–4) and 4 after the pandemic (IQR = 3–6).
Ruiz-Fernández et al. (2020) [95]	To assess compassion fatigue, burnout, compassion satisfaction, and perceived stress in health professionals during the COVID-19 health crisis in Spain.	Spain	*n* = 506 Nurses: 78.7% Physicians: 21.3% Men: 23.3% Women: 76.7%	Burnout, compassion fatigue, and compassion satisfaction (ProQoL-5), and stress (PSS-14): Physicians had higher scores for compassion fatigue (21.6 vs. 19.4, *p* = 0.014) and burnout (26.2 vs. 24.3, *p* = 0.005), compared to nurses. Nurses had higher compassion satisfaction scores (39.9 vs. 37.1, *p* = 0.001) compared to physicians. The perceived stress scores were similar in both professions. Professionals working in specific COVID-19 services and in emergency departments had higher scores for compassion fatigue (*M* = 24.3 ± 8.1) and burnout (*M* = 28.9 ± 7.2).
Sagherian et al. (2020) [91]	To know the prevalence of insomnia, fatigue, and psychological well-being (exhaustion, post-traumatic stress, and psychological distress), and examine the differences in these measures based on the working characteristics of nursing personnel during the COVID-19 pandemic in the USA.	United States of America	*n* = 583 Health professionals: Contact with COVID-19 patients: 277 No contact with COVID-19 patients: 129 Men: 25 Women: 396	Burnout (MBI), compassion fatigue (OFER-15), insomnia (PFI), post-traumatic stress (SPRINT), and psychological distress (PHQ-4): Burnout: emotional exhaustion: *M* = 32.21 ± 12.01 (*n* = 451); depersonalization: *M* = 11.13 ± 6.99 (*n* = 452); personal accomplishment: *M* = 32.95 ± 8.00 (*n* = 450); nurses who cared for COVID-19 patients scored higher for depersonalization (*t* (400) = −2.750, *p* = 0.006). Post-traumatic stress: *M* = 15.32 ± 7.00 (*n* = 502), indicating severe PTSD symptoms; 55.38% (*n* = 278) required an additional clinical evaluation; the nursing staff who cared for COVID-19 patients had higher levels of post-traumatic stress (*t* (402) = −3.276, *p* = 0.001). Insomnia: *M* = 13.50 ± 5.29 (*n* = 564); the nursing staff who cared for COVID-19 patients had greater levels of insomnia (*t* (388) = −2.064, *p* = 0.040). Psychological distress: *M* = 6.10 ± 3.30 (*n* = 422), indicating moderate levels of psychological distress; 47.39% (*n* = 200) of the sample presented possible depression, and 62.32% (*n* = 263) possible anxiety.
Sayilan et al. (2020) [96]	To determine burnout levels and sleep quality in nurses during the COVID-19 pandemic.	Turkey	*n* = 267 nurses Men: 66 Women: 201	Burnout (MBI) and sleep quality (PSQI): Emotional exhaustion (*p* = 0.040) and personal accomplishment (*p* = 0.019) were significantly higher in women. A negative relationship was observed between sleep quality and emotional exhaustion (r = −0.234, *p* < 0.001) and depersonalization (*r* = −0.174, *p* = 0.004), but not with personal accomplishment (*p* > 0.050).
Secosan et al. (2020) [97]	To identify the mediating effect of insomnia and exhaustion on secondary trauma and mental health in frontline health professionals during the COVID-19 pandemic.	Romania	*n* = 126 Nurses: 32 Physicians: 94 Men: 45 Women: 81	Burnout (MBI), secondary traumatic stress (STSS), mental health (MHI-5), and insomnia (ISI): Insomnia was negatively related to exhaustion, (*r* = 0.390, *p* < 0.001) and secondary traumatic stress (β = 0.030; 95% CI [0.000–0.084]). Burnout was significantly positively related to mental health complaints (*r* = 0.560, *p* < 0.001; β = 0.310, *p* < 0.001). Secondary trauma was positively and significantly correlated with insomnia (*r* = 0.59, *p* < 0.001), exhaustion (*r* = 0.470, *p* < 0.001), and mental health complaints (*r* = 0.38, *p* < 0.001). The relationship between insomnia and mental health complaints was partially mediated by secondary traumatic stress and exhaustion.
Soto-Rubio et al. (2020) [98]	To analyze the effect of psychosocial risks and emotional intelligence on burnout.	Spain	*n* = 125 nurses Men: 20.9% Women: 79.1%	Burnout (CESQT), emotional balance (TMMS-24), and psychosocial risks (UNIPSICO): The significant positive predictors of burnout were emotional work (β = 0.160, *p* < 0.050), interpersonal conflict (β = 0.170, *p* < 0.050), role conflict (β = 0.440, *p* < 0.001), and emotional repair (β = 0.260, *p* < 0.001). High emotional attention increased the predictive power of interpersonal conflicts for burnout, while high emotional repair reduced the predictive power of interpersonal conflicts for burnout.
Spiller et al. (2020) [99]	To examine changes in the working hours and mental health of Swiss healthcare workers at the peak of the COVID-19 pandemic, and after flattening of the curve.	Switzerland	*n* = 812 Nurses: 342 Physicians: 470 Men: 232 Women: 580	Burnout (MBI), generalized anxiety (GAD-7), and depression (PHQ-9): Participants in the flattened-infection-rate-curve group reported more exhaustion than those from the time of peak infections (332.5 and 294, respectively, *p* < 0.001, *r* = −0.107). Anxiety among participants from the peak infections group was higher than for those after curve flattening (831.5 and 370, respectively, *p* < 0.001, *r* = 0.125). Both samples presented equal levels of the symptoms of depression (*p* = 0.463).
Tan et al. (2020) [100]	To assess burnout and its associated factors in healthcare professionals.	Singapore	*n* = 3075 Physicians: 458 (14.9%) Nurses: 1394 (45.3%) Others: 483 (15.7%) Men: 794 (25.8%) Women: 2199 (71.5%) Others: 82 (2.7%)	Burnout (OLBI): The mean exhaustion scores were *M* = 2.50, and were higher for nurses (*M* = 2.52). The mean depersonalization scores were highest for administrative staff (*M* = 2.46) and lowest for support staff (*M* = 2.32).
Trumello et al. (2020) [101]	To analyze the psychological adjustment of Italian healthcare professionals during the peak of the COVID-19 pandemic in terms of perceived stress, anxiety, depression, and professional quality of life.	Italy	*n* = 627 Healthcare professionals exposed to COVID-19 patients: 306 Healthcare professionals not exposed to COVID-19 patients: 321	Burnout, compassion fatigue and vicarious trauma (ProQOL-5); anxiety and depression (HADS); and perceived stress (PSS-10): Healthcare professionals who did not work with COVID-19 patients had lower burnout scores (*M* = 26.38 ± 6.76) compared to health workers who worked with COVID-19 patients (*M* = 29.70 ± 7.35; *t* = 24.01, *p* = 0.037). Healthcare professionals exposed to COVID-19 patients showed significantly higher levels of stress (*t* = 8.47, *p* = 0.013), burnout (*t* = 24.01, *p* = 0.037), secondary trauma (*t* = 18.74 *p* = 0.002), anxiety (*t* = 8.59, *p* = 0.014), and depression (*t* = 8.51, *p* = 0.013). No differences in compassion satisfaction were detected. Those working in the Italian regions most affected by the pandemic experienced higher levels of perceived stress (*t* = 7.93, *p* = 0.013) and burnout (*t* = 5.30, *p* = 0.008), and lower levels of compassion satisfaction (*t* = 5.28, *p* = 0.008).
Vagni et al. (2020) [102]	To identify the coping strategies used by emergency and healthcare professionals to deal with stressors related to the COVID-19 pandemic that may be associated with the risk of developing vicarious trauma.	Italy	*n* = 210 Healthcare group: Physicians: 57 (50%) Nurses: 47 (37.3%) Psychologists: 9 (7.14%) Nursing assistants: 7 (5.56%) Emergency group: Firefighters: 21 (23.6%) Civil protection: 20 (22.5%) Men: 90 (42.9%) Women: 120 (57.1%)	Vicarious trauma (STSS-I) and coping with stress (CSES-SF): In the healthcare group, there were significant differences in the effect of stress in decision making (*F* = 3.680; *p* < 0.050), which was higher in physicians (*M* = 14.51 ± 2.89) than psychologists (*M* = 11.11 ± 2.15; *p* < 0.050), as well as for stress due to COVID-19 (*F* = 3.57, *p* < 0.05), which was higher in nurses (*M* = 16.19 ± 3.47) than doctors (*M* = 14.30 ± 3.61; *p* < 0.050). There were no differences in the levels of stress and secondary trauma or coping strategies in the emergency group. Compared to men, women reported greater levels of physical stress (*M* = 10.90 ± 4.83 vs. *M* = 7.30 ± 4.57, *t* = 5.470, *p* < 0.001), higher emotional stress (*M* = 13.30 ± 3.68 vs. *M* = 11.64 ± 3.80, *t* = 3.180, *p* < 0.010), and higher stress caused by COVID-19 (*M* = 14.93 ± 3.68 vs. *M* = 13.58 ± 4.22, *t* = 2.480, *p* < 0.050).
Wahlster et al. (2020) [103]	To assess the concerns of frontline healthcare professionals caring for critically ill COVID-19 patients.	77 countries	*n* = 2700 Physicians: 41% Nurses: 40% Others: 19% Men: 35% Women: 65%	Burnout and emotional distress (ad-hoc): Emotional distress and burnout were significantly associated with female sex (absolute risk reduction [ARR] = 1.16; 95% CI [1.01–1.33]) and being a nurse (ARR = 1.31; 95% CI [1.13–1.53]). Those who had seen 10–50 or >50 COVID-19 patients had a 17% and 28% higher risk of exhaustion than those who had seen <10 patients. The limited availability of PPE and shortage of nurses were associated with a risk of burnout > 30% and 18%, respectively. The most common concerns included transmitting the infection to family members (61%), emotional distress and exhaustion (52%), concerns about their own health (44%), and experiencing social stigma from their communities (21%). All health concerns were highest in North America.
Wu et al. (2020) [104]	To compare the prevalence of burnout in physicians and nurses in frontline and non-frontline services.	United States of America	*n* = 190 Frontline services: Men: 28% Women: 72% Non-frontline services: Men: 6% Women: 94%	Burnout (MBI): the prevalence of exhaustion was significantly lower in the frontline group than in the non-frontline group (13% vs. 39%; *p* < 0.001). The prevalence of a low level of personal accomplishment was lower in the frontline group than in the non-frontline group (39% vs. 61%; *p* = 0.002).
Yörük and Güler (2020) [105]	To investigate the relationship between stress, psychological resilience, burnout syndrome, and sociodemographic factors and depression in midwives and nurses during the COVID-19 outbreak.	Turkey	*n* = 377 Nurses and midwives	Burnout (MBI), perceived stress (PSS), depression (BDI), and resilience (RSA): there was a moderate positive relationship between depression and emotional exhaustion (*r* = 0.490, *p* < 0.001). The risk of depression was 1.92 times higher in midwives than in nurses (95% CI [1.08–3.41]. The mean scores for emotional exhaustion and depersonalization were significantly higher in midwives and nurses with high depression scores, but the personal achievement score was significantly lower in this group. The perceived stress score was significantly higher in midwives and nurses with high depression scores (*p* < 0.001). High psychological resilience was protective against the risk of depression (OR = 0.95; 95% CI [0.93–0.96], *p* < 0.001).
Zerbini et al. (2020) [106]	To explore whether people working in specific COVID-19 services were experiencing greater psychosocial stress compared to their colleagues working in regular services, and whether different healthcare professionals (nurses vs. physicians) were affected differently by the COVID-19 pandemic.	Germany	*n* = 111 Nurses: 75 Nurses working in specific COVID-19 services: 45 Nurses working in regular services: 30 Physicians: 35 Physicians working in specific COVID-19 services: 17 Physicians working in regular services: 18	Burnout (MBI), depression, anxiety, and stress (PHQ): Compared to colleagues working in regular services, nurses working in specific COVID-19 services reported higher levels of burnout (*t* (73) = −2.970, *p* = 0.004) and depressed mood (*t* (73) = −3.066, *p* = 0.003); in addition, their levels of personal accomplishment were lower (*t* (73) = 3.246, *p* = 0.001) compared to physicians. Burnout was related to depression (*r* = 0.550, *p* < 0.001), anxiety (*r* = 0.540, *p* < 0.001), and stress (*r* = 0.540, *p* < 0.001). Comparison between groups did not reveal statistically significant effects.
Zhang et al. (2020) [107]	To identify stressors and burnout in frontline nurses caring for COVID-19 patients in Wuhan and Shanghai, and explore the perceived and effective moral support strategies.	China	*n* = 107 nurses Men: 10 Women: 97	Burnout (MBI) and stress factors (ad- hoc): Burnout: emotional exhaustion: *M* = 12.27 ± 7.14; depersonalization: *M* = 2.07 ± 2.78; personal accomplishment: *M* = 16.44 ± 8.36. Lower age, less work experience, and a longer time spent working in quarantine areas were all associated with higher levels of emotional exhaustion (statistics not specified). The subgroup that had spent the longest time working in quarantined areas had the highest levels of depersonalization (statistics not specified). Stress factors in the COVID-19 questionnaire: homesickness (96.3%), uncertainty about how long the current work status would last (85.0%), concern about infecting oneself (84.1%), skin damage caused by prolonged use of PPE (75.7%), and discomfort caused by PPE (75.7%). Main coping strategies: adopting preventive measures; learning about COVID-19; active learning of professional knowledge; adjusting attitudes and positively facing the COVID-19 epidemic; and talking with family and friends. Main effective supports: support from supervisors; sufficient supplies of material; subsidies provided by the government; clear instructions on treatment procedures; and adequate knowledge of COVID-19.
Zhizhong et al. (2020) [108]	To assess the psychometric properties of the 10-item Moral Injury Symptom Scale for health professionals (MISS-HP).	China	*n* = 3006 Nurses: 583 Physicians: 2423	Burnout (MBI-HSMP), moral injury (MISS-HP), depression (PHQ-9), and generalized anxiety (GAD-7): Emotional exhaustion was correlated with moral damage (*r* = 0.340, *p* < 0.050), depression (*r* = 0.620, *p* < 0.050), anxiety (*r* = 0.620, *p* < 0.050), and well-being (*r* = −0.530, *p* < 0.050). Depersonalization was correlated with moral damage (*r* = 0.400, *p* < 0.050), depression (*r* = 0.590, *p* < 0.050), anxiety (*r* = 0.570, *p* < 0.050), and well-being (*r* = −0.520, *p* < 0.050).

Notes: aMBI = Abbreviated Maslach Burnout Inventory (Maslach et al., 1986); AWS = The Areas of Worklife Scale (Leiter and Maslach, 2004); BDI = Beck Depression Inventory (Beck et al., 1996); BES = Basic Empathy Scale (Jolliffe and Farrington, 2006); BHS = Beck Hopelessness Scale (Beck and Steer, 1993); BRS = Brief Resilience Scale—burnout (Smith et al., 2008); CBI = Copenhagen Burnout Inventory (Kristensen et al., 2005); CD-RISC = Connor–Davidson Resilience Scale 25 (Connor and Davison, 2003); CD-RISC-10 = Connor–Davidson Resilience Scale 10 (Campbell-Stills and Stein, 2010); CESQT = Spanish Burnout Inventory (Gil-Monte et al., 2019); CIS = Coping Inventory for Stressful Situations (Sirigatti, and Stefanile, 2009); CMBI = Chinese version of the Maslach Burnout Inventory (Zhou et al., 2016); CSES-SF = The Coping Self-Efficacy Scale—Short Form (Chesney et al., 2006); DASS-21 = Depression, Anxiety, and Stress Scale 21 (Henry and Crawford, 2005); FCs = short Fatigue Compassion Scale (Adams, 2004); FS-HPs = Fatigue and Fear Scale for Healthcare Professionals (Sakib et al., 2020); GAD-7 = Generalized Anxiety Disorder Assessment–7 (Spitzer, 2006); GSS = Chinese version of the General Self-efficacy Scale (Zhang et al., 1995); HADS = Hospital Anxiety and Depression Scale (Zigmond and Snaith, 1983); IES-6 = Impact of Event Scale—Revised (Weiss, 2007); IES-R = Impact of Event Scale—Revised (Weiss, 1997); ISI = Insomnia Severity Index (Bastien et al., 2001); MBI = Maslach Burnout Inventory (Maslach and Jackson,1996); MBI-HSS = Maslach Burnout Inventory—Human Services Survey (Maslach and Jackson, 1986); MHI-5 = Mental Health Continuum—Short Form (Franken, et al., 2018); MIES = Moral Injury Events Scale (Nash et al., 2013); Mini-Z MBI = Single-Item Maslach Burnout Inventory (Linzer et al., 2016); MISS-HP = Moral Injury Symptoms Scale—Health Professional (Mantri et al., 2020); MSPSS = Multidimensional Scale of Perceived Social Support (Zimet et al., 1988); OFER-15 = Occupational Fatigue Exhaustion Recovery 15 (Winwood, 2006); OLBI = Oldenburg Burnout Inventory (Demerouti et al.,2003); PCL-C = Post-Traumatic Stress Scale (Weathers et al., 1991); PFI = Stanford Professional Fulfillment Index (Trockel, 2018); PHQ 9 = The Patient Health Questionnaire 9 (Kroenke et al., 2001); PHQ = Patient Health Questionnaire (Spitzer et al., 1999); PHQ-2 = The Patient Health Questionnaire 2 (Kroenke et al., 2003); PHQ-4 = The Patient Health Questionnaire 4 (Kroenke et al., 2009); PSQI = Pittsburgh Sleep-Quality Index (Buysse et al., 1989); ProQOL-5 = Professional Quality Of Life Scale version 5 (Stamm et al., 2009); PSS = Perceived Stress Scale (Cohen et al., 1983); PSS-10 = Perceived Stress Scale 10 (Cohen et al., 1983); PSS-14 = Perceived Stress Scale 14 (Cohen et al., 1983); PTGI-SF = Posttraumatic Growth Inventory—Short Form (Tedeschi, and Calhoun, 1996); PWLS = Physician Work–Life Study (Waddimba 2016); QLS = Quality of Life Scale (Menekay and Celmece, 2017); RSA = Resilience Scale for Adults (Friborg et al., 2005); RS-14 = 14-Item Resilience Scale (Wagnild, 2009); SAS = Chinese version of Zung’s Self-Rating Anxiety Scale (Zung 1971); SDS = Chinese version of Zung’s Self-Rating Depression Scale (Zung, 1965); SPRINT = Short Post-Traumatic Stress Disorder Rating Interview (Connor, and Davidson, 2001); STAI = Trait State Anxiety Inventory (Spielberger, 1970); STSQ = Secondary Traumatic Stress Questionnaire (Meda et al., 2012); STSS = Secondary Traumatic Stress Scale (Ting, 2005); STSS-I = Secondary Traumatic Stress Scale Italian Version (Setti and Argentero, 2012); SUD = Scale of Subjective Relief Units (Wolpe, 1990); TMMS-24 = The Trait Meta-Mood Scale (Salovey, 1995); UNIPSICO = Psychosocial Resource Factors (Gil-Monte, 2016); VTS = Vicarious Trauma Survey (Vrklevski and Franklin, 2008); WBI = Well-Being Index (Dyrbye, 2019).

## Data Availability

The data presented in this study are available within the article.

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
