# Peer review of "The Impact of the COVID-19 Pandemic on Burnout, Compassion Fatigue, and Compassion Satisfaction in Healthcare Personnel: A Systematic Review of the Literature Published during the First Year of the Pandemic"

_healthcare, 2022, doi:10.3390/healthcare10020364_

Round 1

Reviewer 1 Report

  1. The review paper is interesting and well written.
  2. The authors reviewed 2,858 records obtained from the CINAHL, Cochrane Library, Embase, PsycINFO, PubMed, and Web of Science databases and included 76 in this review to determine the level of burnout, compassion fatigue, and compassion satisfaction, as well as their associated risk and protective factors, in healthcare professionals during the first year of the COVID-19 pandemic.
  3. The author cites extensive literature to illustrate the research topic.
  4. The content of the research conclusion section is too concise, please reinforce the important conclusions of this study.

Author Response

Point 1: The review paper is interesting and well written.

Response 1: Thank you.

Point 2: The authors reviewed 2,858 records obtained from the CINAHL, Cochrane Library, Embase, PsycINFO, PubMed, and Web of Science databases and included 76 in this review to determine the level of burnout, compassion fatigue, and compassion satisfaction, as well as their associated risk and protective factors, in healthcare professionals during the first year of the COVID-19 pandemic.

Response 2: Thank you.

Point 3: The author cites extensive literature to illustrate the research topic.

Response 3: Thank you.

Point 4: The content of the research conclusion section is too concise, please reinforce the important conclusions of this study.

Response 4: We have developed the conclusions section adapting it to the reviewer's suggestions.

Reviewer 2 Report

I read the paper entitled "The impact of the COVID-19 pandemic on burnout, compassion fatigue, and compassion satisfaction in healthcare personnel: a systematic review of the literature published during the first year of the pandemic". The topic is very interesting and topical.

-However, a problem presented by the manuscript is the time frame in which the job search was made. The pandemic was officially declared by the WHO on March 11, 2020. I would suggest to researchers to add the possible literature from 1 January 2021 to 10 March 2021.

- The questions asked (line 92-97) obviously concern the first year of the pandemic and not the whole pandemic.

-Table 1: Please check the reference in the study by Lange et al [76] is the same as the next Lázaro-Pérez et al [76].

-Line 236-239: I would ask the authors to give the highest and lowest price of burnout presented by the studies.

Author Response

Point 1: I read the paper entitled "The impact of the COVID-19 pandemic on burnout, compassion fatigue, and compassion satisfaction in healthcare personnel: a systematic review of the literature published during the first year of the pandemic". The topic is very interesting and topical.

Response 1: Thank you.

Point 2: However, a problem presented by the manuscript is the time frame in which the job search was made. The pandemic was officially declared by the WHO on March 11, 2020. I would suggest to researchers to add the possible literature from 1 January 2021 to 10 March 2021.

Response 2: Thanks for your comment. The review is adjusted to the year 2020, the fact that some articles were published in the paper version in the year 2021, can cause confusion, but the review was carried out exclusively on the articles published during the year 2020. We needed to close a period of revision to be able to start the reading and analysis. Adding more study time at this point would necessarily mean completely redoing the review. This review offers us a vision of how the pandemic affected in the first months, probably subsequent reviews will offer different results since they evaluate different periods and waves of the pandemic.

Point 3: The questions asked (line 92-97) obviously concern the first year of the pandemic and not the whole pandemic.

Response 3: Totally agree with the reviewer. We have modified the sentence in lines 92-93:

“Specifically, we aimed to answer the following questions all of them referring to the experience lived during the first year of the COVID-19 pandemic:”

Point 4: Table 1: Please check the reference in the study by Lange et al [76] is the same as the next Lázaro-Pérez et al [76].

Response 4: Thank you very much for the comment. We have corrected the mistake.

Point 5: Line 236-239: I would ask the authors to give the highest and lowest price of burnout presented by the studies.

Response 5: Thank you for your comment. We have added the following sentence to fulfill your suggestion:

The highest burnout found in the reviewed studies, has been for infectious disease physicians in the Republic of Korea presenting the 90% of them burnout [88], and the lowest burnout was found in a study carried out in Spain in which burnout was present in 20.4% of health professionals[82].